

# Optimal treatment options for acne scars in patients with historic acne: a systematic review and network meta-analysis

Bingwei Wu, Mingju Gao, Yixuan Zhang and Xinping Bai

Department of Plastic Surgery, The Central Hospital of Wuhan, Tongji Medical College, Huazhong University of Science and Technology, Wuhan, China

## ABSTRACT

**Background.** Acne is a common skin condition that can cause permanent scarring and profoundly affect patients' quality of life. Despite the increasing diversity of acne scar treatments, there is a dearth of comprehensive evidence-based guidelines to help clinicians and patients make the best choices. This study aimed to comprehensively assess the efficacy and safety of existing acne scar treatments through a network meta-analysis.

**Method.** PubMed, Embase, Cochrane Library, and Web of Science were thoroughly searched for relevant studies from database establishment to September 19, 2024. Outcome included Echelle d'evaluation clinique des cicatrices d'acne (ECCA), Goodman and Baron Scale (GBS), pain, patient satisfaction, and adverse events. Bayesian network meta-analyses were performed using the gemtc package in R. Risk of bias was assessed using Cochrane Risk of Bias (RoB 2) tool, while publication bias was assessed *via* funnel plots. The study protocol was registered with PROSPERO (CRD42024598780).

**Results.** A total of 68 randomized controlled trials were enrolled, comprising 4,480 patients with acne scarring. Laser + platelet-rich plasma (PRP) ranked best in reducing ECCA scores (surface under cumulative ranking curve to the total area (SUCRA): 98.4%), laser + filler injection ranked best in reducing GBS (SUCRA: 72.1%), and laser + chemical peels ranked the best in patient satisfaction (SUCRA: 89.6%). Microneedling was ranked as the most tolerable in terms of pain (SUCRA: 72.6%). In addition, no strong evidence suggesting a treatment reduced the risk of erythema nor post-inflammatory hyperpigmentation compared to the other treatments.

**Conclusions.** The evidence suggests laser combined with PRP or filler injections are the best options for reducing scar severity, while laser combined with chemical peeling yields the best patient satisfaction. Laser combined with other therapies should be considered to optimize treatment of acne scarring.

Corresponding author
Xinping Bai, laobai442@qq.com

## INTRODUCTION

Acne is a prevailing chronic inflammatory skin disease with a global prevalence of around 9.4%, predominantly affecting adolescents and young adults (*Reynolds et al.,*

*2024*). Although acne itself is typically temporary, it can leave behind permanent scarring. Recent studies have stated that up to 47% of patients with acne vulgaris develop acne scarring (*Liu et al., 2023*). Not only does acne scarring affect one's appearance, but it can also lead to low self-esteem, social anxiety, and depression (*Tan et al., 2022*). Consequently, it is imperative to provide effective treatments for acne scarring to improve the quality of life of affected patients.

Recent advances in medical technology have enriched the treatment methods for acne scarring, including laser therapy, chemical peels, microneedling, dermal fillers, surgical excision, and platelet-rich plasma (PRP) (*Taub, 2019*). Fractional ablative lasers such as 2,940 nm Erbium Yttrium Aluminium Garnet (Er:YAG) lasers and carbon dioxide (CO2) lasers are the most used treatments for atrophic acne scars (*Paasch et al., 2022*), with most ablative studies reporting 26% to 75% improvements (*Ong & Bashir, 2012*). Chemical peeling is also a traditional treatment for acne scars and its effectiveness has been confirmed in many reports (*O'Connor et al., 2018*; *Sun & Lim, 2023*). Nevertheless, due to the nature of chemical reagents, many patients experience greater pain during the treatment (*Abdel Hay et al., 2016*). Microneedling and PRP are emerging popular options for acne scarring, with high efficacy in cosmetic outcomes, postprocedural downtime, and patient satisfaction (*Hesseler & Shyam, 2019*; *Schoenberg et al., 2020*). In addition, studies have highlighted that the combination of microneedling and PRP is superior in cosmetic results, postoperative recovery, and patient satisfaction compared to microneedling alone (*Chawla, 2014*; *Ibrahim, Ibrahim & Salem, 2018*). Dermal fillers provide a rapid solution for addressing depressed scars; however, periodic reapplication may be required to maintain improved appearance (*Almukhadeb et al., 2023*). For deep or large scars, surgical excision may be required (*Abdel Hay et al., 2016*). These diverse therapeutic modalities allow for personalized treatment plans for patients, while simultaneously presenting clinicians with the challenge of selecting the best treatment option.

Despite many treatment options available, there is still a dearth of comprehensive evidence-based guidelines to help clinicians and patients make the best choice. To our knowledge, only one guideline specifically mentions treatments for acne scarring, although it primarily focuses on acne vulgaris management (*Reynolds et al., 2024*). Current research frequently focuses on the comparison of only two or a few treatment options, which hinders a thorough evaluation of s the relative effectiveness of all available treatment approaches (*Ishfaq et al., 2022*; *Wang et al., 2022*; *Zhang et al., 2022*). This situation emphasizes the urgent requirement for comprehensive investigations that compare the effects of multiple treatments concurrently.

In recent years, several network meta-analyses (NMA) have been conducted to evaluate multiple treatments for acne scarring. For example, one article compared the effectiveness of different laser therapies for acne scars, guiding the clinical choice of laser therapy (*Wang et al., 2023*). A study focused on microneedling and its combination with other therapies, revealing the possible synergistic effects of the combined treatments (*Li, Jia & Zhang, 2024*). Another study compared laser, microneedling, PRP, autologous fat grafting, and their combination therapies, providing preliminary evidence for these emerging modalities (*Jiang et al., 2024*). However, these existing NMAs still have some limitations. First, they

focus on a specific few treatments and fail to comprehensively cover all available treatment options. Second, as new technologies and approaches emerge, some novel treatment modalities may not be included in these studies such as filler injections, subcision, and their combined interventions with other techniques.

More specifically, this NMA aimed to comprehensively assess and compare the efficacy of existing acne scar treatments, including laser therapy, chemical peels, microneedling, dermal fillers, surgical excision, and PRP, using an NMA. This NMA was to systematically review and synthesize existing randomized controlled studies on acne scar treatment in patients with historic acne, assess the relative efficacy and safety, and identify shortcomings in existing studies to point the way for future research.

## MATERIALS & METHODS

An NMA was conducted according to a predefined protocol and reported following the PRISMA 2020 guidelines with a PRISMA checklist which is provided in File S1. The study protocol was registered with PROSPERO (CRD42024598780).

### Data sources and access

PubMed, Embase, Cochrane Library, and Web of Science databases were searched to cover a wide range of relevant literature from database establishment to September 19, 2024. The search strategy utilized the keywords "acne scarring", "laser", "microneedling (MN)", "platelet rich plasma (PRP)", "autologous fat grafting", "drugs", "chemical peels", "minimally invasive surgery", "randomized controlled". The specific search strategies are displayed in File S2.

### Inclusion and exclusion criteria

Inclusion criteria covered: (1) patients with facial acne scars; (2) two or more treatments, such as laser, microneedling, PRP, autologous fat grafting, medications, chemical peels, and minimally invasive surgery; (3) outcome evaluation metrics: the primary endpoints were the scar-related rating scale: Echelle d'evaluation clinique des cicatrices d'acne (ECCA), Goodman and Baron Scale (GBS), and the secondary endpoints were visual analogue scale (VAS), satisfaction rating, and adverse events (AEs) (erythema, edema, pain, and post-inflammatory hyperpigmentation (PIH)); (4) study type: all reports were clinical randomized controlled trials (RCTs).

Articles were excluded if they were meta-analyses, systematic reviews, conference abstracts, letters, guidelines, animal experiments, non-English literature, observational studies, studies lacking complete data, and duplicates.

### Literature selection and data extraction

Two researchers (WB, W and MJ, G) independently screened titles and abstracts and selected literature for inclusion.

Articles included at the title and abstract stage were further screened at the full text level against the inclusion and exclusion criteria, and those that were included were considered for the NMAs. Any disagreements were resolved through discussion or by seeking advice

from a third researcher (XY, Z). When extracting data, a standardized form was utilized to record key information, including authors, publication year, region, study type, sample size, age, sex ratio, scar classification, interventions, follow-up duration, rating scales, and outcome metrics. The outcome metrics included ECCA (*Dreno et al., 2007*), Goodman and Baron Scale (GBS) (*Goodman & Baron, 2006*), VAS (*Yang et al., 2021*), patient satisfaction (*Yang et al., 2021*), and AEs. Any dispute was tackled by a third investigator (YX, Z). To ensure accuracy, all extracted data were input into Excel.

## Assessment of study quality

Two authors used the Cochrane Risk of Bias tool (RoB 2) (*Sterne et al., 2019*) to assess the quality of RCTs in randomized sequence generation, allocation concealment, blinding, missing data, and selective reporting. Each aspect was assessed as "high risk", "low risk", or "some risk". Two authors (WB, W and MJ, G) independently assessed the quality, and a third author (YX, Z) reviewed the discrepancies.

## Data analysis

Synthesized effect sizes were presented as mean differences (MD) and relative risks (RR), depending on the type of outcome, along with the corresponding 95% credible intervals (CrI). The ranking of treatments was determined by calculating the surface under the cumulative ranking curve to the total area (SUCRA) for each intervention. SUCRA is the ratio of the surface under the cumulative ranking curve to the total area of the graph; larger values are indicative of more favorable outcomes (*Salanti, Ades & Ioannidis, 2011*).

Bayesian network meta-analysis was performed using the GeMTC package in the R 4.4.1; A normal likelihood and identity link was applied to continuous outcomes, while a binomial likelihood and log link was applied to binary outcome (*Van Valkenhoef et al., 2012*). Variance scaling factor for the starting values of 2.5 was set and four chains were run, with the initial 10,000 iterations discarded as burn-in, and the results were summarized based on the subsequent 40,000 iterations for each chain with a thinning rate of 1. Convergence was visually assessed using Gelman and Rubin's shrink factor and quantitatively by the potential scale reduction factor (PSFR), where $1 \leq PSRF < 1.05$ was indicative of convergence (*Gelman & Rubin, 1992*). Fixed and random effects models were fitted; the degree of heterogeneity was assessed through the $I^2$ statistic. Results based on the random effects model were presented if $I^2 \geq 50\%$. The priors for the relative effects and between-study standard deviation were determined automatically by GeMTC package. Model fit was determined by mean sum of residual deviance to the number of data points ratio (DPR), with the value of DPR closer to 1 representing a better fit.

Network diagrams were plotted to illustrate comparisons across treatments and to ensure connectivity. Each node represents an intervention, and the node size is proportional to the number of patients who received that intervention (*Chaimani et al., 2013*). Connection lines denote direct comparisons between two interventions, and their width is proportional to the number of trials making that comparison (*Thom et al., 2019*). The node splitting method (*Van Valkenhoef et al., 2016*) was used to check possible inconsistencies when both direct and indirect evidence was available for at least one comparison in the network.

The inconsistency between direct and indirect comparisons was tested by node splitting, a Bayesian $p$-value <0.05 was indicative of inconsistency between the direct and indirect evidence. In addition, sensitivity analyses and publication bias tests were performed. Sensitivity analyses were performed by excluding each article at high risk of bias individually and observing whether the results of the remaining studies appeared significantly different. Funnel plots were created using the network package in STATA 15.1 to assess publication bias (*Egger et al., 1997*). Efficacy or safety was noted when the 95% CrIs excluded the null effect.

## RESULTS

### Literature screening results

A total of 32,404 documents were obtained through database search, and Endnote was utilized to screen the documents. Then, 6,549 duplicates were eliminated, 25,548 documents were eliminated by reading the title abstracts, and 307 documents that met the initial screening were re-screened. Further screening was carried out on study types, subjects, interventions, and outcome indicators. 45 articles were not available in full text, 48 articles were inconsistent with study purpose, 64 articles were inconsistent with the control, six articles were inconsistent with the disease, 65 articles did not report the data required for NMA, four articles were not in English, and one article was a case report. Finally, 68 articles were enrolled. The specific screening process is manifested in Fig. 1.

### Basic characteristics and risk of bias assessment

Basic characteristics of enrolled RCTs are displayed in File S3. A total of 68 RCTs were included, comprising 4,480 patients with acne scarring, of whom approximately 44.34% were male and 55.66% were female, with an average age of 26.34 ± 5.35. 21 investigations were carried out in India, 18 in Egypt, 12 in China, five investigations in Pakistan, five in Iran, two in Korea, and one each in the United States, Japan, Denmark, Brazil, and Israel. In addition, 10 RCTs had three arms and 58 RCTs had two arms.

The risk of bias evaluation is shown in Fig. 2. There were 32 (47.1%) RCTs with a low risk of bias, 29 (42.6%) with an unclear risk of bias, and seven (13.2%) with a high risk of bias (Fig. 2). For sensitivity analyses, studies with high risk were excluded individually to assess changes in effect sizes. The results found no significant changes, indicating that the results were stable.

### NMA results
#### Network diagram

The evidence network diagrams are shown in Figs. 3 and 4. Most of the interventions were compared with laser or microneedling as a control. The sample size of studies in which laser was combined with other treatments (*e.g.*, microneedling, chemical peels, drugs, or filler injections) was relatively low. The NMA results for all outcome measures showed favorable model fit and convergence (Table 1).

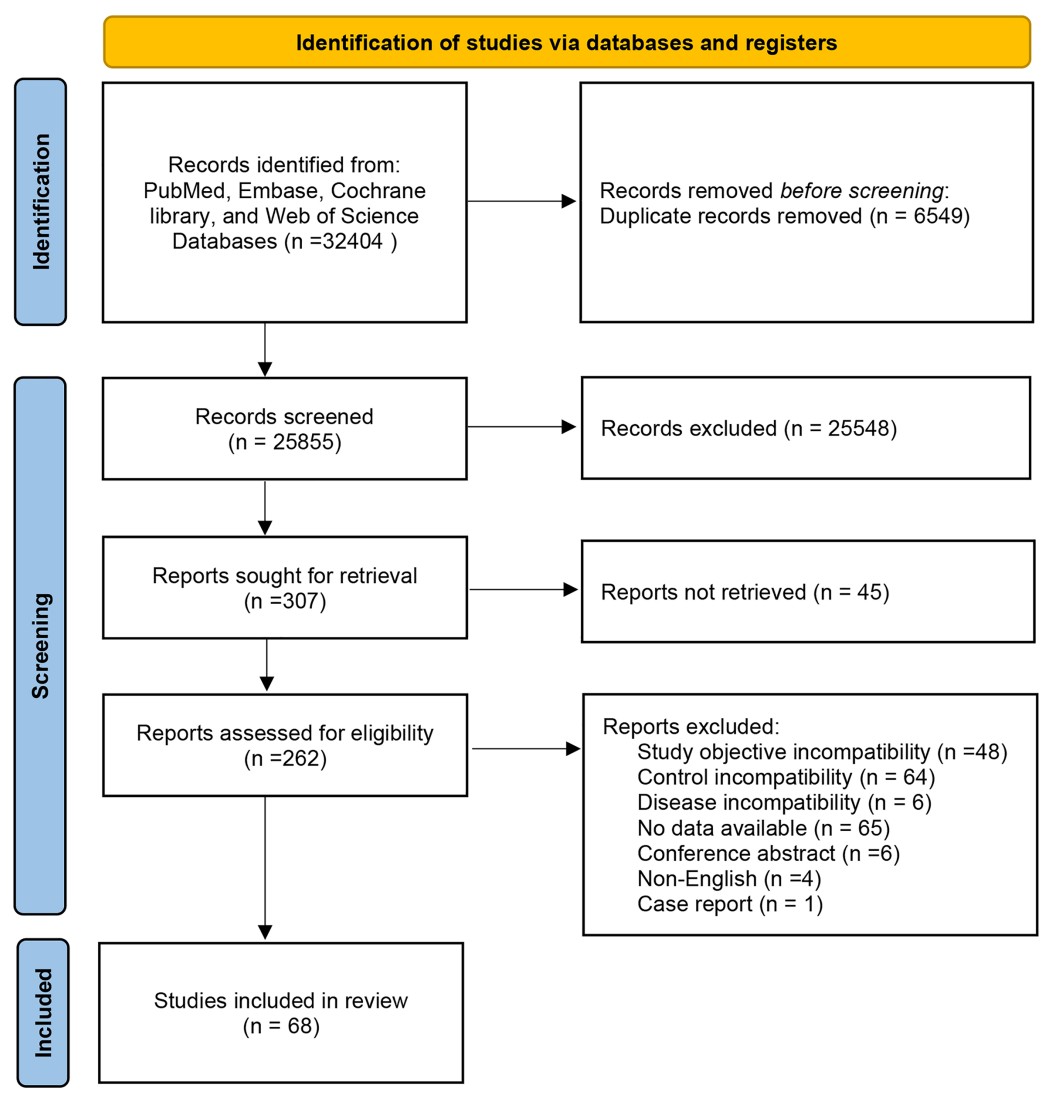

**Figure 1** Flowchart of literature screening.

### ECCA scores

Eighteen RCTs reported ECCA scores. NMA results showed that ECCA scores were notably reduced after laser combined with PRP for acne scarring compared with microneedling, laser, chemical peel, PRP, microneedling + PRP, laser + drugs, and laser + filler injection (laser + PRP $vs.$ microneedling: MD = −21.44, 95% CrI [−27.21 to −15.66]; laser + PRP $vs.$ laser: MD = −20.54, 95% CrI [−21.45 to −19.63]; laser + PRP $vs.$ chemical peeling: MD = −16.66, 95% CrI [−28.44 to −4.75]); laser + PRP $vs.$ PRP: MD = −14.99, 95% CrI [−27.30 to −2.65]; laser + PRP $vs.$ microneedling + PRP: MD = −27.48, 95% CrI [−41.59 to −13.34]; laser + PRP $vs.$ laser + drug: MD = −24.47, 95% CrI [−26.93 to −22.02]; laser + PRP $vs.$ laser + filler injection: MD = −18.55, 95% CrI [−20.25 to −16.86]. More results are detailed in Table 2. SUCRA ranking results revealed laser + PRP

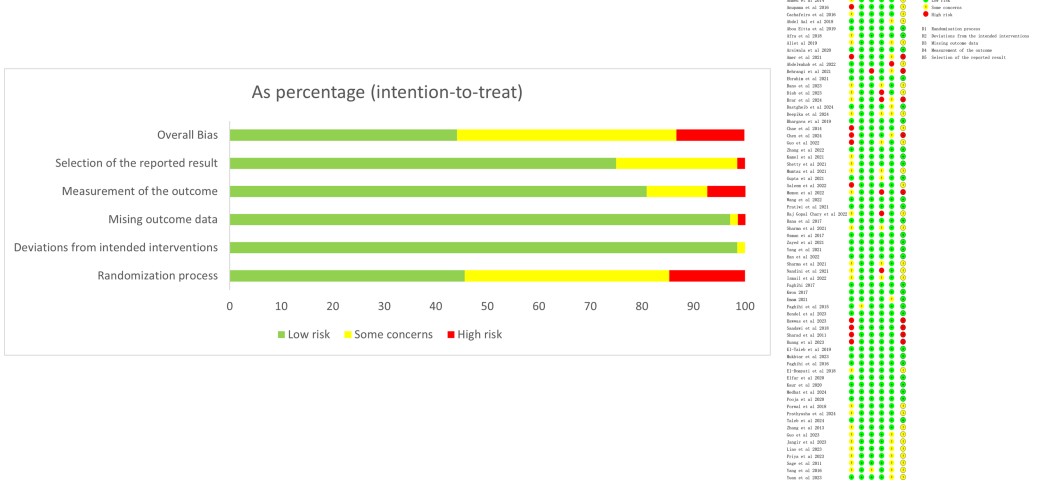

**Figure 2** Risk of bias evaluation.

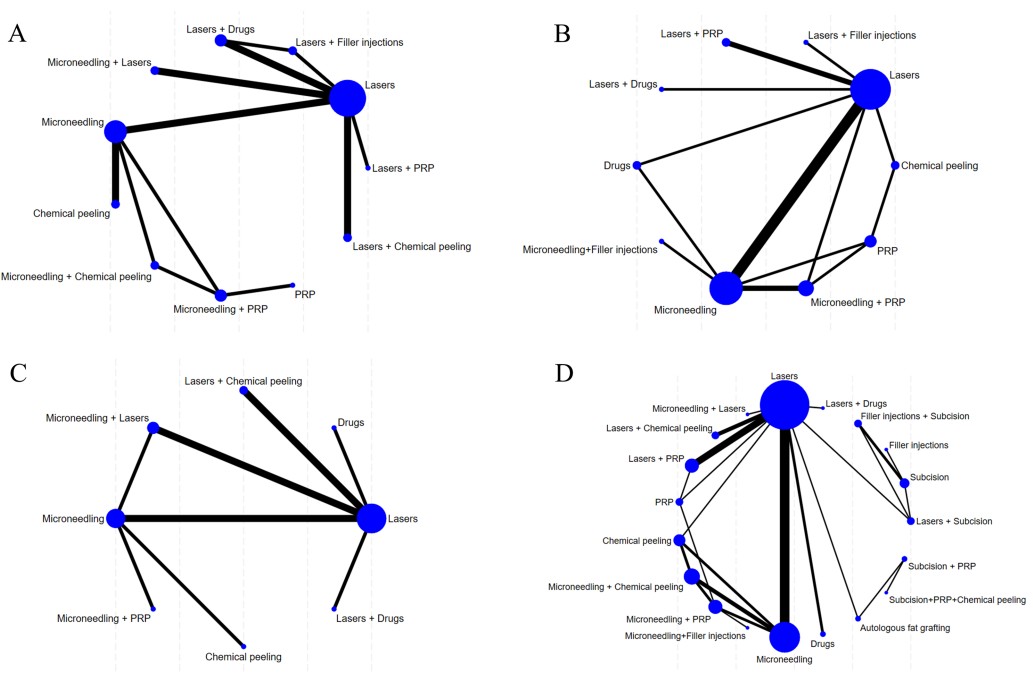

**Figure 3** Network diagram of different outcome indicators. (A) ECCA score; (B) G&B score; (C) VAS score; (D) Patient satisfaction.

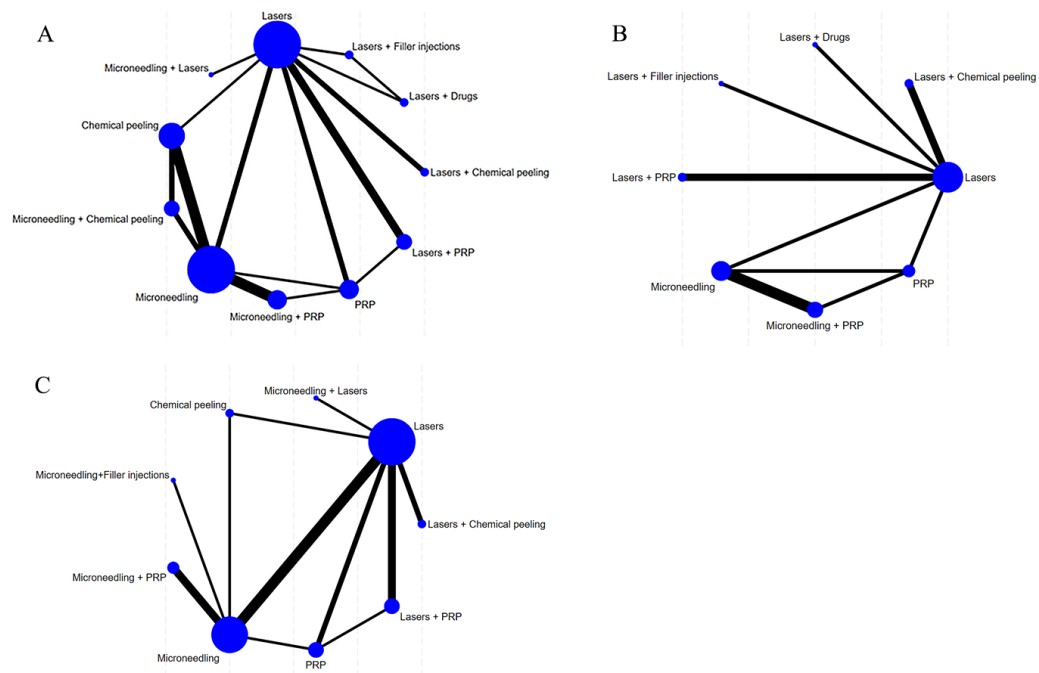

**Figure 4 Network diagram of AEs.** (A) Erythema; (B) Edema; (C) PIH.

**Table 1 NMA heterogeneity, model selection, fitting and convergence results.**

| Outcomes | Overall heterogeneity | | | Model convergence and fit | | | |
|---|---|---|---|---|---|---|---|
| | $I^2(\%)$ | Model | PSRF | DPR | Model | PSRF | DPR |
| ECCA | 40 | Fixed | 1.00 | 1.626 | Random | 1.00 | 1.028 |
| GBS | 68 | Random | 1.00 | 1.010 | Fixed | 1.00 | 3.056 |
| VAS scores | 95 | Random | 1.00 | 1.002 | Fixed | 1.00 | 17.930 |
| Patient satisfaction | 53 | Random | 1.00 | 1.138 | Fixed | 1.00 | 1.515 |
| | | | AEs | | | | |
| Erythema | 10 | Fixed | 1.00 | 1.084 | Random | 1.00 | 0.995 |
| Edema | 5 | Fixed | 1.00 | 1.011 | Random | 1.00 | 1.075 |
| PIH | 87 | Random | 1.00 | 1.003 | Fixed | 1.00 | 1.345 |

**Notes.**
Abbreviations: PSRF, Potential Scale Reduction Factor; DPR, mean sum of residual deviance to the number of data points ratio.

(98.4%) >microneedling + laser (84.2%) >laser + chemical peeling (76.4%). More results are detailed in Fig. 5A.

### GBS

16 RCTs mentioned GBS, and two studies were excluded due to their inability to connect to the network (*Abdelwahab, Omar & Hamdino, 2022*; *Shetty et al., 2021*). NMA results uncovered that the 95% CrIs for comparisons of different regimens contained 0, and it could not imply differences among different regimens in terms of GBS. More results

Wu et al. (2025), *PeerJ*, DOI 10.7717/peerj.19938

Peerj

**Table 2  NMA results of mean difference in change in ECCA score and 95% CrI in different treatment regimens, the summary effect size is the result of comparing the column intervention *vs.* the row intervention, >0 was in favor of row interventions, bolding indicates statistically significant ($p < 0.05$).**

| Microneedling | | | | | | | | | | |
|---|---|---|---|---|---|---|---|---|---|---|
| 0.89 (−4.80, 6.60) | **Lasers** | | | | | | | | | |
| 4.78 (−5.54, 15.16) | 3.88 (−7.88, 15.76) | **Chemical peeling** | | | | | | | | |
| 6.45 (−4.46, 17.37) | 5.55 (−6.73, 17.85) | 1.66 (−13.36, 16.72) | **PRP** | | | | | | | |
| 7.43 (−8.56, 23.51) | 6.53 (−10.43, 23.60) | 2.64 (−16.47, 21.75) | 0.98 (−17.07, 19.03) | **Microneedling+Chemical peeling** | | | | | | |
| −6.05 (−18.92, 6.87) | −6.94 (−21.00, 7.17) | −10.83 (−27.38, 5.73) | −12.50 (−26.29, 1.34) | −13.50 (−29.21, 2.28) | **Microneedling+PRP** | | | | | |
| −3.04 (−9.17, 3.09) | −3.93 (−6.22, −1.66) | −7.81 (−19.89, 4.19) | −9.48 (−22.00, 3.02) | −10.48 (−27.67, 6.69) | 3.01 (−11.29, 17.28) | **Lasers+Drugs** | | | | |
| 2.89 (−3.00, 8.77) | **1.99 (0.56, 3.42)** | −1.89 (−13.84, 9.95) | −3.56 (−15.94, 8.79) | −4.55 (−21.68, 12.52) | 8.94 (−5.25, 23.05) | **5.92 (3.28, 8.58)** | **Lasers+Filler injections** | | | |
| **12.84 (2.03, 23.67)** | **11.96 (2.78, 21.12)** | 8.07 (−6.95, 22.93) | 6.39 (−8.98, 21.73) | 5.40 (−13.93, 24.74) | **18.90 (2.03, 35.73)** | **15.88 (6.44, 25.34)** | **9.96 (0.67, 19.24)** | **Lasers+Chemical peeling** | | |
| **21.44 (15.66, 27.21)** | **20.54 (19.63, 21.45)** | **16.66 (4.75, 28.44)** | **14.99 (2.65, 27.30)** | 14.00 (−3.08, 30.99) | **27.48 (13.34, 41.59)** | **24.47 (22.02, 26.93)** | **18.55 (16.86, 20.25)** | 8.58 (−0.63, 17.80) | **Lasers+PRP** | |
| **15.93 (6.90, 24.98)** | **15.03 (8.04, 22.04)** | 11.16 (−2.66, 24.83) | 9.49 (−4.68, 23.64) | 8.49 (−9.99, 26.87) | **21.98 (6.18, 37.73)** | **18.96 (11.62, 26.36)** | **13.04 (5.89, 20.22)** | 3.07 (−8.46, 14.62) | −5.51 (−12.58, 1.56) | **Microneedling+Lasers** |

**Notes.**

Abbreviations: PRP, platelet-rich plasma.
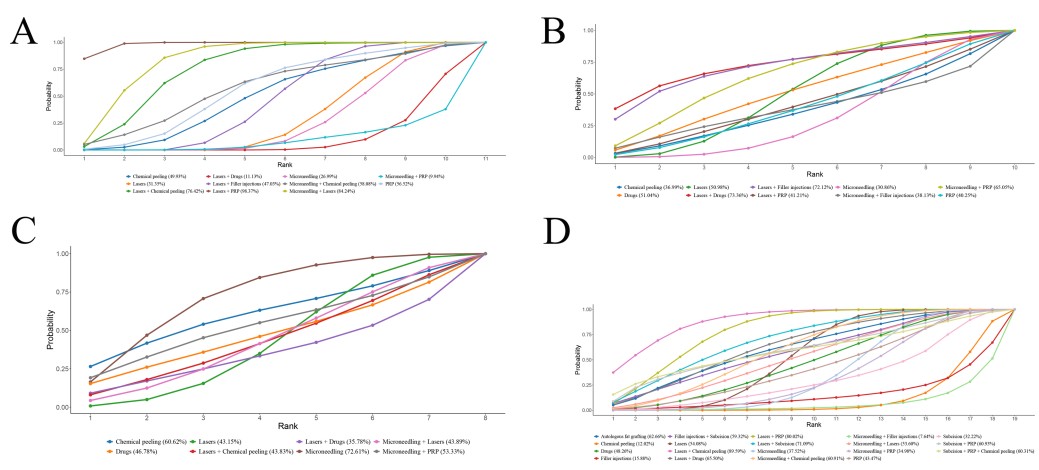

**Figure 5  SUCRA values of different outcomes in different regimens.** (A) ECCA score; (B) G&B score; (C) VAS score; (D) Patient satisfaction.

**Table 3  NMA results of mean difference in change in GBS score and 95% CrI in different treatment regimens, the summary effect size is the result of comparing the column intervention *vs.* the row intervention, >0 was in favor of row interventions, bolding indicates statistically significant (*p* < 0.05).**

| **Microneedling** | | | | | | | | | |
|---|---|---|---|---|---|---|---|---|---|
| 1.69 (−2.8, 6.39) | **Lasers** | | | | | | | | |
| 0.21 (−8.33, 8.98) | −1.47 (−9.81, 6.88) | **Chemical peeling** | | | | | | | |
| 0.64 (−6.38, 7.83) | −1.04 (−8.71, 6.59) | 0.43 (−7.77, 8.6) | **PRP** | | | | | | |
| 1.79 (−5.72, 9.44) | 0.11 (−7.52, 7.65) | 1.59 (−9.45, 12.39) | 1.15 (−8.99, 11.23) | **Drugs** | | | | | |
| 3.17 (−2.91, 9.42) | 1.48 (−5.35, 8.34) | 2.94 (−6.62, 12.53) | 2.52 (−5.02, 10.09) | 1.38 (−8.06, 10.89) | **Microneedling+PRP** | | | | |
| 5.56 (−6.77, 18.05) | 3.86 (−7.7, 15.38) | 5.35 (−8.96, 19.51) | 4.92 (−8.99, 18.73) | 3.75 (−10.07, 17.6) | 2.38 (−11.07, 15.72) | **Lasers+Drugs** | | | |
| 4.95 (−6.26, 16.15) | 3.25 (−7.03, 13.33) | 4.71 (−8.48, 17.9) | 4.29 (−8.55, 17) | 3.15 (−9.58, 15.83) | 1.77 (−10.67, 13.96) | −0.64 (−16, 14.71) | **Lasers+Filler injections** | | |
| 0.74 (−7.71, 9.33) | −0.93 (−8.11, 6.12) | 0.54 (−10.48, 11.52) | 0.1 (−10.43, 10.51) | −1.05 (−11.48, 9.37) | −2.42 (−12.36, 7.39) | −4.81 (−18.34, 8.78) | −4.18 (−16.6, 8.3) | **Lasers+PRP** | |
| 0.06 (−10.71, 10.66) | −1.64 (−13.41, 9.84) | −0.17 (−14.04, 13.48) | −0.61 (−13.55, 12.18) | −1.72 (−14.96, 11.24) | −3.1 (−15.61, 9.09) | −5.49 (−21.89, 10.8) | −4.88 (−20.34, 10.53) | −0.69 (−14.47, 12.83) | **Microneedling+Filler injections** |

**Notes.**

Abbreviations: PRP, platelet-rich plasma.

are detailed in Table 3. SUCRA ranking results revealed laser + drugs (73.36%) >laser + filler injections (72.1%) >microneedling + PRP (65.1%). More results are detailed in Fig. 5B. Due to the lack of evidence to support differences in treatment efficacy among interventions, SUCRA results were informative only and cannot be effectively interpreted as differences in efficacy among interventions.

### Pain based on VAS scores

A total of 11 RCTs reported pain based on VAS scores, and one study was excluded due to their inability to connect to the network (*Dastgheib et al., 2024*). NMA results uncovered that the 95% CrIs for comparisons of different regimens contained 0, and it could not imply differences among different regimens in terms of VAS scores. More results are detailed in Table 4. SUCRA ranking results revealed microneedling (72.6%) >chemical peeling (60.6%) >microneedling + PRP (53.3%). More results are detailed in Fig. 5C. Due

**Table 4** NMA results of mean difference in change in VAS score and 95% CrI scores in different regimens, the summary effect size is the result of comparing the column intervention *vs.* the row intervention, >0 was in favor of row interventions, bolding indicates statistically significant (*p* < 0.05).

| | | | | | | | |
|---|---|---|---|---|---|---|---|
| **Microneedling** | | | | | | | |
| −3.05 (−9.87, 3.89) | **Lasers** | | | | | | |
| −0.78 (−11.94, 10.47) | 2.28 (−10.91, 15.46) | **Chemical peeling** | | | | | |
| −2.85 (−15.95, 10.46) | 0.2 (−11.08, 11.49) | −2.06 (−19.34, 15.26) | **Drugs** | | | | |
| −1.82 (−13, 9.37) | 1.23 (−12.03, 14.35) | −1.05 (−16.87, 14.69) | 1.04 (−16.41, 18.33) | **Microneedling+PRP** | | | |
| −4.38 (−17.49, 8.91) | −1.35 (−12.58, 9.91) | −3.6 (−20.93, 13.73) | −1.54 (−17.44, 14.48) | −2.58 (−19.86, 14.83) | **Lasers+Drugs** | | |
| −3.13 (−13.61, 7.38) | −0.09 (−7.96, 7.82) | −2.36 (−17.75, 13.02) | −0.28 (−14.03, 13.48) | −1.31 (−16.66, 14.05) | 1.24 (−12.5, 14.98) | **Lasers+Chemical peeling** | |
| −3 (−10.96, 5.03) | 0.04 (−6.83, 6.96) | −2.22 (−16.09, 11.56) | −0.15 (−13.49, 13) | −1.18 (−14.83, 12.63) | 1.38 (−11.8, 14.53) | 0.13 (−10.35, 10.61) | **Microneedling+Lasers** |

**Notes.**

Abbreviations: PRP, platelet-rich plasma.

to the lack of evidence to support differences in treatment efficacy among interventions, SUCRA results were informative only and cannot be effectively interpreted as differences in tolerability among interventions.

### Patient satisfaction

A total of 33 RCTs reported post-treatment patient satisfaction grading, and this outcome was analyzed by comparing the proportions of patients that were satisfied or very satisfied and the proportion of patients that were not. NMA results uncovered that patient satisfaction with laser + chemical peeling for acne scarring was considerably better than microneedling alone, laser, chemical peeling, and microneedling + PRP (laser + chemical peeling *vs.* microneedling: RR = 2.04, 95% CrI [1.23–3.85]; laser + chemical peeling *vs.* laser: RR = 1.71, 95% CrI [1.09–2.93]; laser + chemical peeling *vs.* chemical peeling: RR = 3.56, 95% CrI [1.79–7.97]; laser + chemical peeling *vs.* microneedling + PRP: RR = 2.15, 95% CrI [1.15–4.94]), suggesting that laser + chemical peeling was superior for acne scarring treatment. More results are detailed in Table 5. SUCRA ranking results revealed lasers + chemical peeling (85.6%) >lasers + PRP (80.0%) >lasers + subcision (71.1%). More results are detailed in Fig. 5D.

### AEs

*Erythema.* A total of 21 RCTs reported the incidence of erythema after treatment. NMA noted that the 95% CrIs of the comparisons of different regimens included 1, and it could not be assumed a difference in erythema incidence among different regimens. More results are detailed in Table 6. SUCRA ranking results revealed laser + drugs (84.4%) >PRP (72.5%) >microneedling + chemical peeling (65.0%). More results are detailed in Fig. 6A.

*Edema.* A total of 11 RCTs reported the incidence of post-treatment edema. NMA showed that edema incidence of edema after PRP alone was notably lower than that of laser + chemical peeling (PRP *vs.* laser + chemical peeling: RR = 0.06, 95% CrI [0–0.73]) (*P* < 0.05). The 95% CrI of other comparisons contained 1, and it could not yet imply

**Table 5  NMA results of relative risks and 95% CrI of patient satisfaction in different regimens, the summary effect size is the result of comparing the column intervention *vs.* the row intervention, <1 was in favor of row interventions, bolding indicates statistically significant (*p* < 0.05).**

| 1 | 2 | 3 | 4 | 5 | 6 | 7 | 8 | 9 | 10 | 11 | 12 | 13 | 14 | 15 | 16 | 17 | 18 | 19 |
|---|---|---|---|---|---|---|---|---|---|---|---|---|---|---|---|---|---|---|
| **Microneedling** | | | | | | | | | | | | | | | | | | |
| 0.84 (0.62, 1.08) | **Lasers** | | | | | | | | | | | | | | | | | |
| **1.73 (1.03, 3.02)** | **2.07 (1.23, 3.72)** | **Chemical peeling** | | | | | | | | | | | | | | | | |
| 0.97 (0.44, 1.96) | 1.15 (0.56, 2.29) | 0.56 (0.22, 1.28) | **PRP** | | | | | | | | | | | | | | | |
| 0.90 (0.48, 1.75) | 1.07 (0.62, 2.00) | 0.52 (0.23, 1.16) | 0.93 (0.39, 2.5) | **Drugs** | | | | | | | | | | | | | | |
| 1.95 (0.47, 8.66) | 2.33 (0.59, 10.24) | 1.12 (0.25, 5.36) | 2.01 (0.44, 10.68) | 2.17 (0.47, 10.33) | **Filler injections** | | | | | | | | | | | | | |
| 0.73 (0.28, 1.79) | 0.87 (0.35, 2.08) | 0.42 (0.14, 1.15) | 0.75 (0.25, 2.38) | 0.81 (0.27, 2.25) | 0.37 (0.07, 1.91) | **Autologous fat grafting** | | | | | | | | | | | | |
| 0.78 (0.51, 1.18) | 0.93 (0.58, 1.54) | **0.45 (0.25, 0.78)** | 0.80 (0.37, 1.90) | 0.87 (0.40, 1.81) | 0.40 (0.09, 1.76) | 1.06 (0.40, 2.99) | **Microneedling+ Chemical peeling** | | | | | | | | | | | |
| 1.05 (0.70, 1.73) | 1.25 (0.80, 2.25) | 0.61 (0.32, 1.20) | 1.09 (0.52, 2.64) | 1.17 (0.55, 2.60) | 0.54 (0.12, 2.48) | 1.44 (0.55, 4.28) | 1.36 (0.87, 2.27) | **Microneedling+ PRP** | | | | | | | | | | |
| 0.72 (0.34, 1.44) | 0.86 (0.44, 1.67) | **0.41 (0.17, 0.95)** | 0.74 (0.29, 2.03) | 0.80 (0.31, 1.86) | 0.37 (0.07, 1.70) | 0.98 (0.33, 3.00) | 0.92 (0.39, 2.05) | 0.68 (0.27, 1.49) | **Lasers+Drugs** | | | | | | | | | |
| 0.49 (0.26, 0.81) | **0.58 (0.34, 0.91)** | **0.28 (0.13, 0.56)** | 0.5 (0.22, 1.17) | 0.54 (0.23, 1.10) | 0.25 (0.05, 1.05) | 0.67 (0.24, 1.81) | 0.63 (0.29, 1.19) | **0.46 (0.20, 0.87)** | 0.68 (0.28, 1.49) | **Lasers+Chemical peeling** | | | | | | | | |
| 0.60 (0.37, 0.90) | **0.72 (0.49, 0.99)** | **0.35 (0.17, 0.64)** | 0.62 (0.31, 1.27) | 0.67 (0.31, 1.26) | 0.31 (0.07, 1.26) | 0.82 (0.32, 2.13) | 0.78 (0.41, 1.34) | **0.57 (0.28, 0.98)** | 0.85 (0.38, 1.73) | 1.23 (0.69, 2.27) | **Lasers+PRP** | | | | | | | |
| 0.84 (0.39, 1.73) | 1.00 (0.5, 2.00) | 0.48 (0.19, 1.14) | 0.86 (0.34, 2.42) | 0.94 (0.36, 2.23) | 0.43 (0.08, 2.01) | 1.15 (0.38, 3.55) | 1.08 (0.45, 2.45) | 0.8 (0.31, 1.78) | 1.17 (0.45, 3.05) | 1.71 (0.77, 4.19) | 1.39 (0.66, 3.12) | **Microneedling+ Lasers** | | | | | | |
| 1.18 (0.41, 3.36) | 1.41 (0.52, 3.94) | 0.68 (0.21, 2.14) | 1.21 (0.38, 4.43) | 1.32 (0.40, 4.15) | 0.61 (0.20, 1.58) | 1.62 (0.43, 6.38) | 1.52 (0.49, 4.62) | 1.12 (0.34, 3.35) | 1.64 (0.50, 5.62) | 2.41 (0.82, 7.94) | 1.95 (0.69, 6.00) | 1.40 (0.42, 4.91) | **Subcision** | | | | | |
| 0.78 (0.27, 2.13) | 0.93 (0.35, 2.50) | 0.45 (0.14, 1.36) | 0.8 (0.25, 2.82) | 0.87 (0.27, 2.64) | 0.40 (0.12, 1.19) | 1.07 (0.29, 4.06) | 1.00 (0.33, 2.95) | 0.74 (0.23, 2.13) | 1.09 (0.33, 3.58) | 1.59 (0.55, 5.05) | 1.29 (0.47, 3.81) | 0.93 (0.28, 3.11) | 0.66 (0.38, 1.11) | **Filler injections+ Subcision** | | | | |
| 0.67 (0.29, 1.44) | 0.80 (0.38, 1.66) | 0.38 (0.15, 0.94) | 0.69 (0.26, 1.98) | 0.75 (0.28, 1.83) | 0.34 (0.09, 1.10) | 0.91 (0.29, 2.93) | 0.86 (0.35, 2.03) | 0.63 (0.24, 1.47) | 0.93 (0.34, 2.51) | 1.36 (0.58, 3.45) | 1.11 (0.50, 2.58) | 0.79 (0.29, 2.19) | 0.57 (0.28, 1.11) | 0.86 (0.44, 1.63) | **Lasers+ Subcision** | | | |
| 2.44 (0.89, 7.54) | 2.92 (1.04, 9.42) | 1.4 (0.46, 4.76) | 2.53 (0.79, 9.85) | 2.72 (0.82, 9.8) | 1.25 (0.21, 7.77) | 3.36 (0.88, 14.73) | **3.14 (1.13, 9.79)** | 2.3 (0.91, 6.33) | **3.41 (1.01, 13.24)** | **5.02 (1.65, 18.53)** | **4.06 (1.40, 14.15)** | 2.91 (0.86, 11.53) | 2.07 (0.49, 9.74) | 3.14 (0.77, 14.63) | **3.66 (1.05, 14.82)** | **Microneedling+ Filler injections** | | |
| 0.73 (0.20, 2.59) | 0.87 (0.24, 3.05) | 0.42 (0.1, 1.62) | 0.75 (0.18, 3.27) | 0.81 (0.19, 3.14) | 0.37 (0.05, 2.4) | 1.00 (0.41, 2.48) | 0.94 (0.24, 3.53) | 0.69 (0.17, 2.57) | 1.02 (0.24, 4.18) | 1.5 (0.40, 5.94) | 1.21 (0.33, 4.56) | 0.87 (0.21, 3.65) | 0.62 (0.12, 3.04) | 0.94 (0.19, 4.58) | 1.09 (0.25, 4.69) | 0.30 (0.05, 1.49) | **Subcision+PRP** | |
| 0.73 (0.17, 3.06) | 0.87 (0.21, 3.61) | 0.42 (0.09, 1.89) | 0.75 (0.16, 3.81) | 0.81 (0.17, 3.66) | 0.37 (0.05, 2.68) | 1.00 (0.33, 3.09) | 0.94 (0.20, 4.14) | 0.69 (0.14, 2.99) | 1.02 (0.21, 4.87) | 1.49 (0.34, 6.99) | 1.21 (0.28, 5.40) | 0.87 (0.18, 4.24) | 0.62 (0.11, 3.47) | 0.94 (0.16, 5.24) | 1.09 (0.22, 5.45) | 0.30 (0.05, 1.69) | 1.00 (0.51, 1.97) | **Subcision+PRP+ Chemical peeling** |

**Notes.**

Abbreviations: PRP, platelet-rich plasma.

**Table 6  NMA results of relative risks and 95% CrI of erythema in different regimens, the summary effect size is the result of comparing the column intervention *vs.* the row intervention, >1 was in favor of row interventions, bolding indicates statistically significant ($p < 0.05$).**

| Microneedling | | | | | | | | | | |
|---|---|---|---|---|---|---|---|---|---|---|
| 0.56 (0.27, 1.20) | Lasers | | | | | | | | | |
| 0.98 (0.67, 1.59) | 1.76 (0.77, 3.96) | Chemical peeling | | | | | | | | |
| 1.15 (0.64, 2.28) | 2.06 (0.95, 4.60) | 1.18 (0.57, 2.49) | PRP | | | | | | | |
| 1.09 (0.41, 3.37) | 1.93 (0.57, 7.51) | 1.10 (0.40, 3.36) | 0.94 (0.30, 3.31) | Microneedling + Chemical peeling | | | | | | |
| 1.00 (0.72, 1.41) | 1.79 (0.80, 3.79) | 1.03 (0.56, 1.67) | 0.87 (0.46, 1.50) | 0.93 (0.28, 2.60) | Microneedling + PRP | | | | | |
| 1.69 (0.39, 8.29) | 2.96 (0.86, 12.36) | 1.71 (0.37, 8.70) | 1.46 (0.32, 7.22) | 1.55 (0.24, 9.86) | 1.68 (0.38, 8.33) | Lasers + Drugs | | | | |
| 1.07 (0.25, 5.16) | 1.86 (0.57, 7.74) | 1.08 (0.24, 5.29) | 0.92 (0.21, 4.50) | 0.98 (0.15, 6.12) | 1.06 (0.25, 5.23) | 0.63 (0.28, 1.42) | Lasers + Filler injections | | | |
| 0.58 (0.25, 1.55) | 1.03 (0.68, 1.84) | 0.59 (0.24, 1.61) | 0.50 (0.21, 1.32) | 0.54 (0.13, 2.07) | 0.58 (0.25, 1.60) | 0.35 (0.08, 1.38) | 0.56 (0.13, 2.05) | Lasers + Chemical peeling | | |
| 0.80 (0.35, 1.92) | 1.42 (0.94, 2.20) | 0.81 (0.32, 2.04) | 0.69 (0.28, 1.66) | 0.73 (0.18, 2.68) | 0.79 (0.34, 1.97) | 0.48 (0.11, 1.82) | 0.76 (0.17, 2.76) | 1.37 (0.67, 2.49) | Lasers + PRP | |
| 0.26 (0.03, 1.56) | 0.46 (0.06, 2.39) | 0.26 (0.03, 1.61) | 0.22 (0.03, 1.36) | 0.23 (0.02, 1.79) | 0.26 (0.03, 1.57) | 0.15 (0.01, 1.28) | 0.24 (0.02, 1.98) | 0.44 (0.05, 2.42) | 0.32 (0.04, 1.76) | Microneedling + Lasers |

**Notes.**

Abbreviations: PRP, platelet-rich plasma.

**Table 7  NMA results of relative risks and 95% CrI of edema in different regimens, the summary effect size is the result of comparing the column intervention *vs.* the row intervention, >1 was in favor of row interventions, bolding indicates statistically significant ($p < 0.05$).**

| Microneedling | | | | | | | |
|---|---|---|---|---|---|---|---|
| **0.31 (0.11, 0.76)** | Lasers | | | | | | |
| 1.11 (0.58, 2.2) | **3.59 (1.47, 9.99)** | PRP | | | | | |
| 1.12 (0.76, 1.69) | **3.62 (1.45, 10.3)** | 1.01 (0.55, 1.8) | Microneedling + PRP | | | | |
| **0.07 (0, 0.81)** | 0.24 (0.01, 2.15) | **0.06 (0, 0.72)** | **0.06 (0, 0.73)** | Lasers+Chemical peeling | | | |
| 0.73 (0.05, 26.29) | 2.36 (0.19, 76.21) | 0.66 (0.04, 23.3) | 0.65 (0.04, 23.25) | 11.2 (0.33, 1,196.19) | Lasers + Filler injections | | |
| **0.31 (0.11, 0.81)** | 1.01 (0.74, 1.38) | **0.28 (0.1, 0.73)** | **0.28 (0.09, 0.73)** | 4.21 (0.46, 127.69) | 0.43 (0.01, 5.49) | Chemical peeling | |
| 0.42 (0.14, 1.19) | 1.37 (0.85, 2.24) | 0.38 (0.12, 1.06) | 0.38 (0.12, 1.07) | 5.74 (0.61, 178.15) | 0.58 (0.02, 7.73) | 1.36 (0.77, 2.42) | Lasers + PRP |

**Notes.**

Abbreviations: PRP, platelet-rich plasma.

differences in edema incidence among different regimens. More results are detailed in Table 7.

SUCRA ranking results revealed PRP (82.2%) >microneedling + PRP (77.6%) >microneedling (69.4%). More results are detailed in Fig. 6B.

*PIH.* A total of 16 RCTs reported the incidence of post-treatment PIH. NMA results uncovered that the 95% CrIs for comparisons of different regimens contained 1, and it could not imply differences among different regimens in terms of PIH incidence. More results are detailed in Table 8. SUCRA ranking results revealed microneedling (72.0%) >microneedling + filler injection (61.4%) >microneedling + PRP (58.4%). More results are detailed in Fig. 6C. Due to the lack of evidence to support differences in treatment efficacy between interventions, SUCRA results were informative only and cannot be effectively interpreted as differences in PIH among interventions.

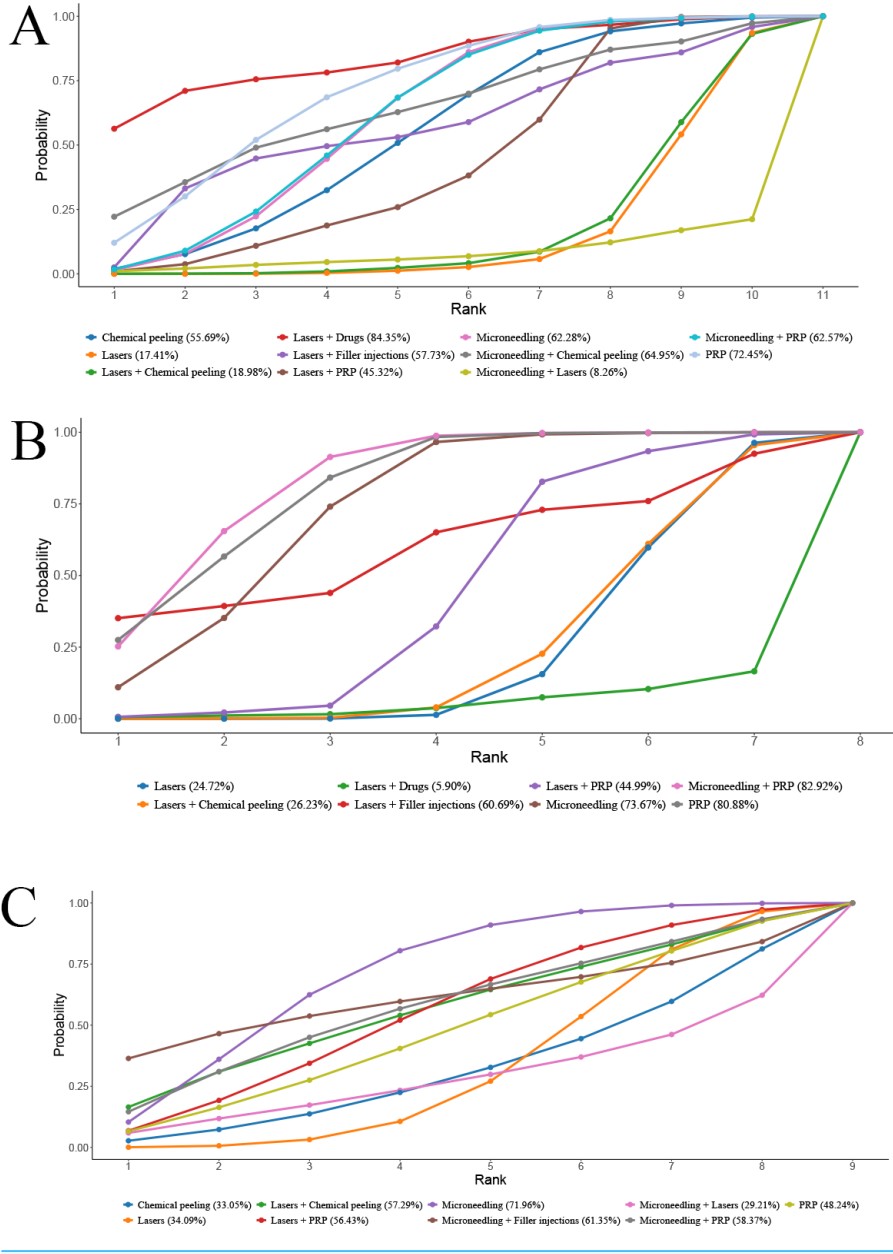

**Figure 6** **SUCRA values of different AEs in different regimens.** (A) Erythema; (B) Edema; (C) PIH.

### Publication bias

A correction-comparison funnel plot was drawn by performing publication bias tests on included indicators. The results were largely symmetrical from left to right, suggesting a low likelihood of publication bias (Figs. 7–8).

## DISCUSSION

Due to advances in equipment and technology in recent years, various options are now available for treating acne scars, mainly including lasers, radiofrequency, microneedling,

**Table 8 NMA results of relative risks and 95% CrI of PIH in different regimens, the summary effect size is the result of comparing the column intervention *vs.* the row intervention, >1 was in favor of row interventions, bolding indicates statistically significant ($p < 0.05$).**

| Microneedling | | | | | | | | |
|---|---|---|---|---|---|---|---|---|
| 0.39 (0.09, 1.50) | **Lasers** | | | | | | | |
| 0.34 (0.05, 2.66) | 0.86 (0.13, 6.88) | **Chemical peeling** | | | | | | |
| 0.54 (0.06, 4.58) | 1.35 (0.24, 8.88) | 1.59 (0.11, 20.92) | **PRP** | | | | | |
| 0.76 (0.11, 5.00) | 1.94 (0.18, 21.02) | 2.23 (0.13, 33.98) | 1.42 (0.08, 24.58) | **Microneedling + PRP** | | | | |
| 0.71 (0.05, 9.83) | 1.81 (0.22, 17.52) | 2.11 (0.11, 39.15) | 1.34 (0.08, 23.18) | 0.95 (0.04, 24.73) | **Lasers + Chemical peeling** | | | |
| 0.66 (0.08, 4.22) | 1.68 (0.40, 6.52) | 1.97 (0.15, 19.09) | 1.23 (0.15, 8.39) | 0.86 (0.05, 12.84) | 0.92 (0.06, 11.41) | **Lasers + PRP** | | |
| 0.24 (0.01, 5.73) | 0.63 (0.03, 11.01) | 0.72 (0.02, 21.73) | 0.46 (0.01, 13.17) | 0.32 (0.01, 13.22) | 0.34 (0.01, 12.3) | 0.37 (0.01, 9.37) | **Microneedling + Lasers** | |
| 1.01 (0.02, 68.25) | 2.61 (0.03, 227.63) | 2.99 (0.03, 301.63) | 1.90 (0.02, 209) | 1.36 (0.01, 134.76) | 1.43 (0.01, 202.96) | 1.56 (0.02, 170.84) | 4.21 (0.02, 885.86) | **Microneedling + Filler injections** |

**Notes.**

Abbreviations: PRP, platelet-rich plasma.

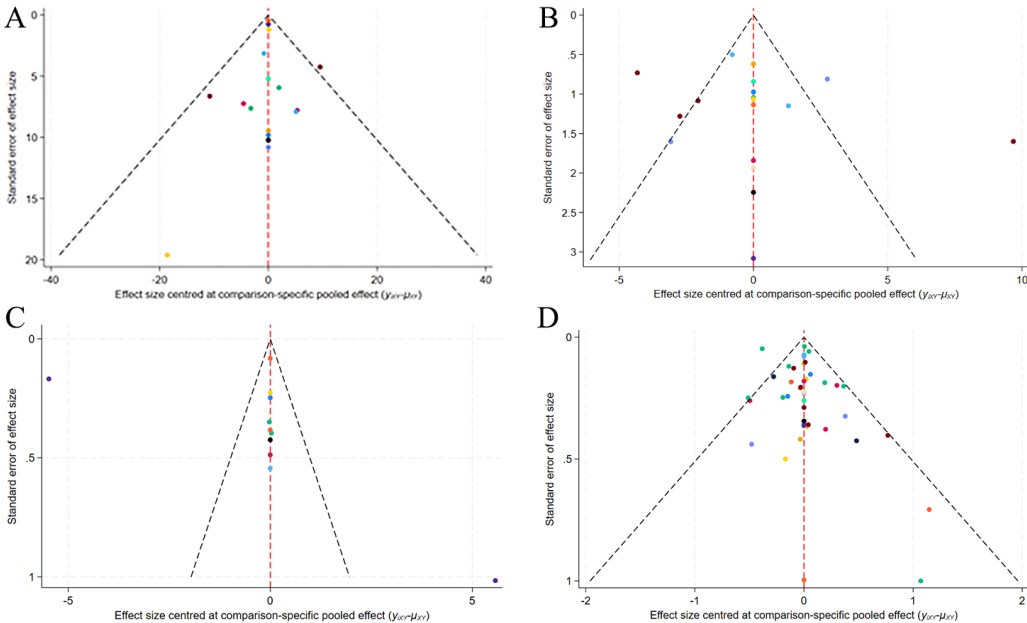

**Figure 7 Network diagram of different outcome indicators.** (A) ECCA score; (B) G&B score; (C) VAS score; (D) Patient satisfaction.

microdermabrasion, chemical peeling, dermal fillers, and surgical techniques, previously published studies have shown the effectiveness of these treatments compared to usual care (*Al-Talib et al., 2017*; *Qoreishi et al., 2025*; *Shen et al., 2022*; *Zaleski-Larsen et al., 2016*). Most previous meta-analyses on acne scarring compared drugs or laser therapy alone (*Wang et al., 2023*). The current NMA compares the efficacy and safety of different treatments for acne scarring, including laser, chemical peeling, and microneedling treatments, which were not well represented in older evidence syntheses.

Laser + PRP ranked best in reducing ECCA scores, and was notably more efficacious than microneedling alone, lasers alone, chemical peeling, PRP alone, microneedling +

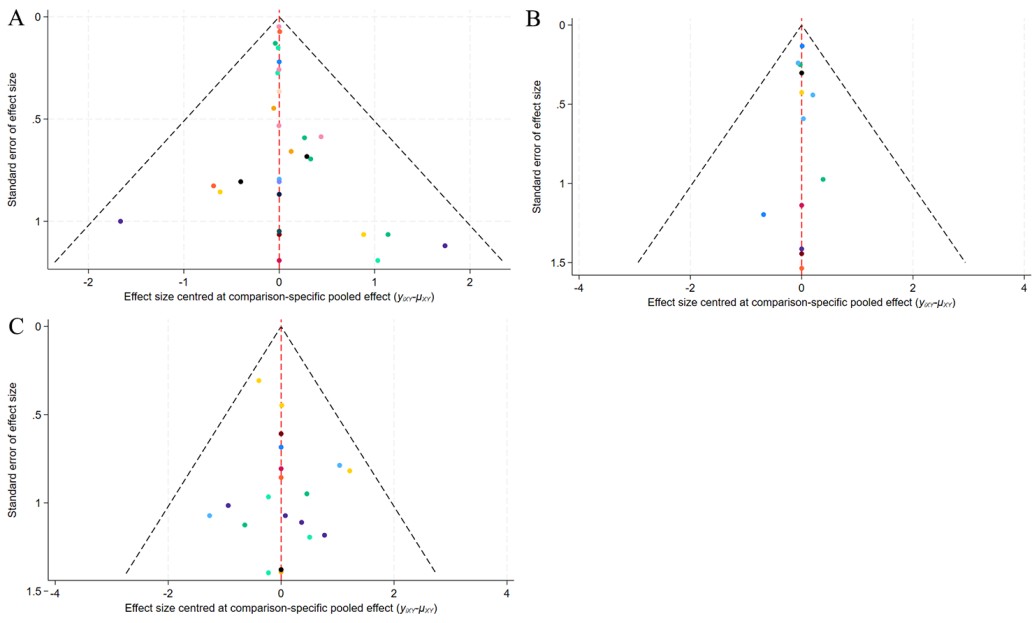

**Figure 8  Funnel plot of AEs.** (A) Erythema; (B) Edema; (C) PIH.

PRP, lasers + drugs, lasers + filler injections. Laser + filler injection ranked best in terms of improving GBS, but there was no evidence to conclude a notable difference in efficacy between treatments. Microneedling ranked best in terms of yielding the least pain measured by VAS scores, but there was no evidence to conclude a notable difference in tolerability between treatments. Laser + chemical peeling ranked best in patient satisfaction, and was notably more efficacious than microneedling alone, lasers alone, chemical peeling alone, and microneedling + filler injections. Additionally, in terms of AEs, the incidence of edema, after laser + chemical peeling for acne scarring was higher than that of PRP alone, and the rate of PIH and erythema after microneedling + lasers for acne scarring was higher than that of others, although there was not enough evidence to conclude a notable difference for any comparison.

Laser + PRP and laser + filler injection have shown significant efficacy in improving acne scarring (*Wu et al., 2021*). Current lasers for scar treatment include $CO_2$ fractional laser and Erbium fractional laser, which are characterized by narrower pulses and higher energies. They combine fractional photothermolysis to vaporize the local tissues of skin lesions and make many microholes in the skin deep into the dermis uniformly, stimulating the production of collagen in the dermis and the rearrangement of collagen fibers to improve the scar (*Lanoue & Goldenberg, 2015*). PRP is a concentrate of autologous platelets, which are 3.0 to 6.4 times more abundant than blood, are rich in alpha granules, and release diverse growth factors that can stimulate collagen and healing (*Bariya et al., 2011*). In areas of tissue injury, platelets are the first cells to reach the site and release growth factors that are crucial for tissue repair (*Gawdat et al., 2014*). Previous articles have demonstrated the healing-promoting and anti-inflammatory properties of PRP (*Harris, Naidoo & Murrell,*

*2015*), which is effective in both repairing damaged nerve tissue and relieving pain. The combination of laser and PRP produces additive effects that result in maximum efficiency, consistent with previous findings. A conventional meta-analysis by *Aljefri et al. (2022)* demonstrated that laser + PRP was highly synergistic, effective, and safe in the treatment of moderate to severe atrophic acne scars.

Filler injection refers to the local injection of non-permanent fillers (hyaluronic acid, collagen), permanent fillers (silicone, silicone, poly-L-lactic acid), and autologous component grafts (fat grafts, autologous fibroblasts grafts) to fill in soft tissue defects, eliminate wrinkles, and other cosmetic purposes. Fillers can increase the volume of tissue in depressed scars and stimulate fibroblasts to produce collagen. Hyaluronic acid dressing contains abundant hyaluronic acid, which is the primary component of the extracellular matrix, binds to the keratinocyte-forming cell receptor CD44 at the wound, stimulates cell proliferation and migration, and promotes wound healing, and its water-holding capacity allows for a favorable microenvironment for wound healing (*Zhang et al., 2023*).

Microneedling has shown favorable results in relieving pain compared to other measures. Invented by Dr. Horst in France, microneedling therapy uses microscopic needle-like instruments to puncture the skin to create orifices that activate the skin's self-injury repair mechanism to promote regeneration of fibers, dermis, and epithelial cells (*Shen et al., 2022*). Besides, microscopic tubes are created in the skin for a short period, facilitating the passage of cell growth factors and other such repair components, which penetrate directly into the deeper layers of the skin, accelerating the formation of epithelial tissues and increasing the rate of healing (*Park et al., 2010*). Studies have reported that microneedling treatment resulted in increased expression of type I collagen, and important signaling molecules for collagenogenesis and neovascularization such as glycosaminoglycans, vascular endothelial growth factor, fibroblast growth factor-7, epidermal growth factor, and TGF-$\beta$ (*Busch et al., 2018*; *Moftah et al., 2018*).

Patient satisfaction with laser + chemical peels is higher than other treatments. Chemical peels have the effect of reducing stratum corneum thickness, loosening intercellular adhesion in the stratum corneum, and inhibiting acne formation (*Kontochristopoulos & Platsidaki, 2017*). Furthermore, chemical peels can improve atrophic acne scars by stimulating dermal fibroblasts to proliferate and produce new collagen (*Puri, 2015*). *Fabbrocini et al. (2010)* reported that chemical reconstruction of skin scars used a high concentration of trichloroacetic acid to thicken the dermis and produce collagen, thus minimizing the side effects of scarring and PIH.

In a comparative consideration of various treatments, laser + PRP or filler injection showed the best efficacy in improving acne scarring, but they can be more painful, whereas microneedling and chemical peels are appropriate for patients with less severe conditions or those seeking less pain and lower risk of complications.

In the treatment of acne scarring, different modalities are suitable for different types of scarring and patient needs, and choosing the right method is crucial. The biggest advantage of laser treatment is that it is highly targeted and can precisely affect the dermis layer, which is effective for depressed scarring and has a short recovery period. Microneedling stimulates the skin's self-healing mechanism to improve scarring, is relatively simple to

perform, well tolerated, and has a short recovery period (*Han et al., 2025*). In contrast, chemical peel surgery is less costly and widely applicable. Filler injection can quickly fill the depressed area, with immediate and noticeable effects. The treatment needs to be selected in conjunction with the scar type, individual differences, and economic situation.

This NMA provides an important reference for the clinical treatment of acne scarring. Through a coherent comparison of multiple treatment options, patients can rationally choose the treatment option with the best efficacy and safety profiles, thus developing a more personalized acne scar treatment plan for them. With more in-depth research on the efficacy of acne scarring treatments, new modalities that are more effective and safer may be developed in the future to further enhance patients' quality of life and treatment experience. This NMA attempts to discuss current therapeutic tools and their effectiveness in clinical practice, in the hope of providing more clinical references and guidance for acne scarring treatment. It is also recommended that clinicians should update their knowledge, pay attention to new technologies, and try various combinations of treatment options to achieve the optimal management of acne scarring. This study also has some notable limitations. First, although NMA can synthesize results from multiple articles, it is still limited by the quality and consistency of data from the included articles. Heterogeneity remains a potential problem, as baseline characteristics of participants, details of treatment regimens, and duration of follow-up may vary across studies. Second, some literature was not included in this NMA due to patient unblinding or allocation concealment, which would impact the overall quality assessment. Third, subgroup analyses were not performed to determine differences among populations and regions, thus failing to give the best treatment suggestions for different populations. Finally, the GBS score was regarded as a continuous variable, which was necessary for the reporting format of the included studies but was also a potential limitation. Future trials should report categorical frequency distributions to allow for more detailed sequential analyses. In addition, the fragmented nature of current evidence on reporting demographic variables limits the exploration of modifiers of treatment efficacy. A standardized framework for patient data collection and sharing is a key prerequisite for the precision treatment of acne scarring.

## CONCLUSION

This NMA provides evidence suggesting that laser combined with other therapies should be considered to optimize treatment of acne scarring. More evidence is required to conclude definitive superiority of these treatments over all other regimens, and to better understand tolerability as well as the risks of adverse events.

### Funding
The authors received no funding for this work.

### Competing Interests
The authors declare there are no competing interests.

## Author Contributions

- Bingwei Wu conceived and designed the experiments, performed the experiments, analyzed the data, prepared figures and/or tables, authored or reviewed drafts of the article, and approved the final draft.
- Mingju Gao conceived and designed the experiments, analyzed the data, authored or reviewed drafts of the article, and approved the final draft.
- Yixuan Zhang conceived and designed the experiments, analyzed the data, prepared figures and/or tables, authored or reviewed drafts of the article, and approved the final draft.
- Xinping Bai analyzed the data, authored or reviewed drafts of the article, and approved the final draft.

## Data Availability

Data is available in the Supplemental Files.

## Supplemental Information

Supplemental information for this article can be found online at http://dx.doi.org/10.7717/peerj.19938#supplemental-information.

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
