# Peer review of "Optimal treatment options for acne scars in patients with historic acne: a systematic review and network meta-analysis"

_PeerJ, doi:10.7717/peerj.19938_

## Round 0.1 · original submission · Major Revisions

Your manuscript has been reviewed and requires several modifications prior to making a decision. The comments of the reviewers are included at the bottom of this letter. The reviewers acknowledge the manuscript’s clear language, comprehensive literature review, and robust methodology employing Bayesian network meta-analysis. However, they highlight key areas for improvement:

(1) Statistical methods require further clarification, including sensitivity analyses, heterogeneity assessment, and model fit reporting.

(2) The dataset should be shared for transparency and reproducibility.

(3) The analysis of ordinal data should be revised, as certain scales may not be appropriate for continuous treatment.

(4) Subgroup analyses based on patient demographics are recommended to enhance clinical relevance.

Additional refinements in figures, tables, and result interpretations would further strengthen the manuscript.

Reviewer 1 ·

Basic reporting

The manuscript is written in clear and professional English, with terminology appropriate for an academic audience. The introduction provides sufficient context and justification for the study, outlining the prevalence of acne scars and the need for an evidence-based approach to treatment selection.

The literature review is comprehensive and well-referenced, drawing on relevant prior studies. The structure follows standard scientific formatting, including clear headings for methodology, results, and discussion. Figures and tables are well-structured, clearly labeled, and provide relevant data. Additionally, raw data has been shared, contributing to transparency and reproducibility.

However, minor linguistic improvements could enhance readability. A thorough proofreading to refine sentence structures and remove any ambiguity is recommended.

Experimental design

The study aligns well with the aims and scope of the journal, presenting original research in the form of a network meta-analysis. The research question is clearly defined and addresses an important knowledge gap in dermatological treatment strategies.

The methodology is robust, employing Bayesian network analysis in R software and assessing publication bias using statistical methods in StataMP. The inclusion and exclusion criteria are clearly outlined, ensuring methodological rigor. The choice of databases (PubMed, Embase, Cochrane Library, Web of Science) strengthens the reliability of the study, and the PRISMA checklist is included, ensuring adherence to systematic review standards.

One potential area for improvement is the explanation of statistical methods. While Bayesian network models are appropriate for this type of analysis, additional clarification regarding sensitivity analyses and how inconsistency in the data was addressed would strengthen the reproducibility of the findings.

Validity of the findings

The findings are well-supported by the provided data, and all relevant statistical analyses have been conducted to ensure robustness. The conclusions align with the original research questions and are limited to supporting results. The study appropriately refrains from assessing the impact and novelty of the findings, maintaining a focus on statistical soundness.

The use of 68 randomized controlled trials (RCTs) covering 4,480 patients strengthens the reliability of the study. Furthermore, all underlying data appear to be statistically sound and controlled. The use of SUCRA rankings effectively differentiates between treatment options.

One potential limitation is the lack of subgroup analysis based on patient demographics. Including insights into how age, gender, or skin type might influence treatment efficacy would enhance the clinical applicability of the results.

Additional comments

This study provides a valuable synthesis of existing acne scar treatments using a rigorous methodological approach. While the findings are robust, minor refinements in the explanation of statistical methods and additional subgroup analyses could enhance its impact. The manuscript is well-prepared for publication after these minor revisions.

Reviewer 2 ·

Excellent Review

This review has been rated excellent by staff (in the top 15% of reviews)
EDITOR COMMENT
This review represents exceptional detail, objectivity, and constructiveness, providing clear and actionable recommendations to improve the manuscript. The reviewer has thoroughly assessed the transparency and reproducibility of the study, emphasizing the need for comprehensive model reporting, dataset availability, and clarity in statistical methods. Their critique of the network meta-analysis approach, particularly regarding heterogeneity modeling, outcome categorization, and effect size calculations, is highly insightful and supported by relevant literature. Importantly, the reviewer has not only identified major methodological concerns but also provided specific guidance on how to address them, such as improving model fit reporting, refining the treatment categorization, and enhancing result interpretation. Their engagement with the broader literature strengthens the review’s impact, offering valuable context for the study’s contribution to the field. This detailed and constructive feedback will significantly enhance the manuscript’s clarity, rigor, and clinical relevance. We sincerely appreciate the reviewer’s time, expertise, and dedication to high-quality peer review.

Basic reporting

Major: Sufficiency of model reporting
As a reader, I would expect to see enough information that I would be able to replicate the analysis. For example, the authors have specified the main software and version, to which should be added detail on packages used (e.g. gemtc) and any additional information on priors. The authors do mention the number of chains and iterations, which is exactly what is needed for reproducibility, on page 8, line 127.
The authors provide no convergence statistics or model fit. There is mention of heterogeneity in the discussion but no discussion of model structure, an estimate of the extent of heterogeneity or inconsistency in the methods or results. Prior to fitting node-split models, I would expect to see the model fit reporting global inconsistency.
It would be helpful for the reader if the authors could add detail within the methods section on the model fitting process, the prior distribution given to the parameter capturing heterogeneity (if used) and clarify the meaning of annealings (page 8, line 128) in this context.
In the results it would be helpful to see a table setting out model fit for the chosen models, as well as fit statistics for other models that were fitted but discounted: e.g. a comparison of the overall NMA model and the node-split models. If heterogeneity was explored, it is necessary to report fit statistics from i) models that do not include heterogeneity in treatment effects arising from study-level differences (sometimes termed common or fixed effect) and ii) models including a heterogeneity parameter (random effect) for comparison.
Major: Dataset not provided
The authors note that there is no requirement to provide the data because it is extracted from published studies. I disagree with this assessment and would encourage the authors to provide their dataset using a link to a repository or as a csv/text file. I have found OSF to be user-friendly for this purpose. I suggest this because i) it allows the authors’ work to be verified by other groups, supporting reproducibility and the authors’ future work; ii) reviewing the dataset can provide the reader with answers to seemingly anomalous results, e.g. large uncertainty around the estimates of relative risks for treatments where there were few events.
Minor: Background and context
In order to understand the potential for heterogeneity and the comparability of treatment options, it would be helpful to the reader of a general journal (such as PeerJ) to specify whether all of these treatments are offered at the same point in the clinical pathway. E.g. are some most effective during active inflammation and others in the years after? Is the patient population reflected here still experiencing active acne or are these residual scars? The patient population (mild/severe/previous acne) could be specified in the title.
This work estimates treatment differences between newer interventions that appear not to have been included in past systematic reviews. It would be worth mentioning other existing NMAs to ensure that the introduction properly represents the current literature: for example Stuart et al. (2021), Mavranezouli et al. (2022), and Harper et al. (2024).
The title needs to contain more information to make it useful and specific. Since the main focus of this review was the comparison of mechanical and laser treatments, this should be reflected in the title, as should the presence of the systematic review (consistent with PRISMA guidelines).
In the materials and methods, the authors summarise the outcomes clearly. Given that this is a general journal rather than one specific to acne, it would be helpful to briefly summarise the suitability of these outcomes in capturing scarring, and providing references for all scales used. References for the ECCA and GBS scales are easily found but the VAS and satisfaction scales used for acne are not easily located and so cannot be missing.

Experimental design

Major: Choice of model for outcome
In several cases, it appears that the outcome is ordinal, with a small number of discrete categories representing levels of severity. In particular, the GBS score cannot be analysed as a continuous scale but should be analysed as ordered categorical data. Agresti (2010) contains an introduction to the analysis of ordinal data and Morris et al. (2024) may be of interest if studies in this area report GBS scores with some contraction of the 4-point scale; they also link to R code. The Technical Support Documents in the Evidence Synthesis series (Dias et al. 2016) contain WinBUGS code for the meta-analysis of ordinal data that may be of interest to the authors if they are more comfortable with BUGS.

It may be possible to analyse the VAS scale as continuous, depending on the length of the scale and reporting limitations in the papers. There should be confirmation of whether these scores are patient-reported or clinician-reported, since previous work on these outcomes suggests that the two may differ systematically, with patients reporting different scores to clinicians.

On line 124, the authors specify that they analysed standardised mean differences (SMDs). Ordinarily, these would only be used where the evidence is reported on different scales in different studies, requiring the analyst to standardise the data before synthesis. If SMDs were used here, it would be helpful to describe – in supplementary material if necessary – how these were calculated. Higgins et al. (2023) contains details on how SMDs are calculated in Cochrane evidence syntheses. The effect sizes for ECCA, GBS and VAS are extremely high if on the SMD scale, therefore this should be clarified.

Major: Extent of the treatment set
The intervention labelled ‘drugs’ in the network plots should be updated to specify the treatment received: e.g. “topical and oral pharmacological treatments”. I suggest that the authors provide context for the very small number of studies (or possibly participants) where ‘drugs’ were trialled, given that previous evidence syntheses suggest that these are widely used treatment options. For example, was the evidence set separated by severity? If the authors were focusing on the population with extremely severe acne, or scarring following historic acne, it is possible that certain interventions would be excluded as unsuitable treatment options, but this is not clearly described within the paper currently.

Validity of the findings

Major:
1. The authors have conducted network meta-analysis and node-splitting models but it is not clear why the node-splitting models were conducted (was there a prior belief of conflict between direct and indirect evidence?) or how these results are interesting. Further comment on the meaning of the highlighting in the tables may clear this up, otherwise I would suggest that the node-splitting results could be moved to the supplementary material if they are unremarkable.
2. Page 10 (PDF version), line 229, the credible interval indicated is enormous, suggesting either that the model has not converged or that there is no information on the rate of oedema for this treatment comparison (e.g. there were zero events on all arms of all studies informing the treatment comparison). If the model that was fitted included heterogeneity (i.e. was a random-effects model estimating between-study variation), I would encourage the authors to check to see whether there is enough evidence to estimate the between-study variation. It may only be possible to fit a fixed-effect model if there are few edges in the network informed by more than one study.
3. Page 10 (PDF version), line 238, the authors report SUCRA scores for three treatments: microneedling, microneedling + filler and microneedling + PRP. The order of effects and the presence of microneedling + filler suggests that this text refers to panel B, not C. It looks as though the labelling of plot panels (B and C) has been reversed.

Minor:
4. Page 11, line 250, the current text is too strong. I’d suggest the following edit. From, “The current NMA first compares the efficacy and safety of all treatments for acne scarring, including laser, drugs, and chemical peeling.” To, “The current NMA compares the efficacy and safety of treatments for acne scarring, including laser, chemical peeling and microneedling treatments not well represented in older evidence syntheses.”
5. Page 12, lines 297-298, the last sentence of this paragraph is not supported by the findings contained here and should be removed: this work does not demonstrate microneedling’s simplicity, overall safety or long-term effects.
6. One problem often encountered by evidence synthesists is how best to present the estimates of clinical effectiveness, particularly when there are multiple outcomes of interest. The authors have chosen to present the treatment effects as central estimates and credible intervals in tables (sometimes called mileage charts or triangle tables). This is an excellent choice for presenting a large quantity of information but it is difficult to pick out the most interesting results and can be time-consuming to reach an overview of which treatments are effective for particular outcomes. Forest plots are often used for this purpose. If the authors would prefer not to include forest plots, it would be helpful to the reader if an additional sentence could be added to the results in each section to confirm how key treatments perform relative to a reference treatment (e.g. “lasers”).
7. Tables need more detail to be clear: i.e. “Mean (or median) difference in change in ECCA score and 95% CI (or CrI)”
8. What estimates are in the upper and lower triangles of the mileage charts: is it network estimates and an average estimate from node-splitting models? Need text explaining direction of dominance if this flips between upper and lower triangles.
9. In page 10 (PDF version), line 221, the authors refer to hypothesis testing and the 95% CI – these reflect the language of frequentist statistics. I’d expect to see the authors refer to credible intervals (CrI)
10. Page 10 (PDF version), line 238, where there is no evidence to support a difference in treatment effects between the interventions because of high uncertainty, there should be a note that SUCRA scores can be difficult to interpret.
11. The authors use rankings and effect estimates throughout, skilfully bringing together two different types of result. Treatment effects provide evidence on which treatments work at all, and rankings provide an order. Within the discussion, there should be further mention of which treatments are effective, since it is not clear to the reader whether the set of ranked treatments are all effective (though some more than others), all ineffective, or some effective and some ineffective.
12. Figures 2 & 3 – Odd gridlines behind network plots
13. Figures 5 & 6 – Colours are not sufficiently easy to discriminate. It is possible that combining colour and line type would allow the reader to distinguish between lines more easily. However, in panel 5d it may be necessary to label the lines with numbers and provide a key in order to distinguish lines.
14. Figure 7 – again, there are too many values for colour alone to be helpful in discriminating points. The authors could use numbers rather than colours, or only label points of interest on the edge of the range.
15. In list of figures (p13 of manuscript, p18 of PDF) Figure 7 is mislabelled – label identical to Fig 3.
16. Values given to different number of significant figures in the tables.
17. Colour highlighting of cells not explained.
18. Selection of Subject Terms: I’d suggest that the authors reconsider the subject terms selected: ‘dermatology’ is suitable, but ‘data mining and machine learning’ or ‘healthcare services’ do not reflect the work reported here. The key words chosen seem appropriate.

Additional comments

This manuscript represents a great deal of work. The authors screened a large number of papers and report their searches in appropriate detail. The authors have included a large number of outcomes and evidently intend to capture the effect of treatment on both the physical manifestations of acne and the emotional toll.

I note four major comments: i) the overall reporting of the methodology; ii) provision of the dataset; iii) the selection of statistical models, analysing ordinal data as continuous and continuous outcomes as SMDs; and iv) the exclusion of topical and oral pharmacological treatments from the evidence set. I also provide several, more minor comments, including the authors’ review of the existing literature and the justification of the outcomes chosen.

References:
Agresti A. Analysis of ordinal categorical data / Alan Agresti. 2nd ed. Wiley series in probability and statistics; 2010.
Dias S, Welton NJ, Sutton AJ & Ades AE. NICE DSU Technical Support Document 2: A Generalised Linear Modelling Framework for Pairwise and Network Meta-Analysis of Randomised Controlled Trials. 2011; last updated September 2016; available from http://www.nicedsu.org.uk
Harper JC, Baldwin H, Choudhury SP, Rai D, Ghosh B, Aman MS, Choudhury AR, Dutta SK, Dey D, Bhattacharyya S, Lin T, Joseph G, Dashputre AA, Tan JKL. Treatments for Moderate-to-Severe Acne Vulgaris: A Systematic Review and Network Meta-analysis. J Drugs Dermatol. 2024 Apr 1;23(4):216-226.
Higgins JPT, Li T, Deeks JJ (editors). Chapter 6: Choosing effect measures and computing estimates of effect [last updated August 2023]. In: Higgins JPT, Thomas J, Chandler J, Cumpston M, Li T, Page MJ, Welch VA (editors). Cochrane Handbook for Systematic Reviews of Interventions version 6.5. Cochrane, 2024. Available from www.training.cochrane.org/handbook.
Mavranezouli I, Daly CH, Welton NJ, Deshpande S, Berg L, Bromham N, Arnold S, Phillippo DM, Wilcock J, Xu J, Ravenscroft JC, Wood D, Rafiq M, Fou L, Dworzynski K, Healy E. A systematic review and network meta-analysis of topical pharmacological, oral pharmacological, physical and combined treatments for acne vulgaris. Br J Dermatol. 2022 Nov;187(5):639-649.
Morris P, Wang C, O'Connor A. Network meta-analysis for an ordinal outcome when outcome categorization varies across trials. Syst Rev. 2024 May 9;13(1):128.

Reviewer 3 ·

Basic reporting

More care is needed to adhere to the PRISMA 2020 checklist. For example:
1) Two different versions of the PRISMA checklist are provided. The PRISMA 2020 checklist should only be presented and followed.
2) Location of where an item is reported: It looks like only the page numbers of the broad section (e.g., Introduction, Methods, Results, Discussion) are reported. This needs to be more specific. Section numbers would be helpful. Figures and tables may be cited as well.
3) Title: “Systematic review” should be included in the title.
4) Abstract: The PRISMA 2020 checklist advises users to refer to the “PRISMA 2020 for Abstracts” checklist. The current abstract does not include information on the study inclusion/exclusion criteria, the date of the database searches, limitations, and protocol registration number. In addition, summary estimates and confidence/credible intervals should be presented (not just SUCRA). Finally, the software packages should be stated (e.g., gemtc package in R, network graphs package in STATA), not just the software program.
5) Introduction: The objectives need to be explicitly stated (e.g., in terms of the patient population, interventions, outcomes, and study design).
6) Methods – Search strategy: The specific dates of the database searches need to be reported.
7) Methods – Selection process: The screening process should be addressed in a separate section from “Data extraction”. It is not clear what process was used for screening the full text, and how disagreements were resolved during the screening process.
8) Methods – Synthesis methods: No information was reported on data preparation (Item 13b), nor any methods to explore heterogeneity (Item 13e). If no methods were used, state this.
9) Results – Study selection: No information is provided on the specific studies that appeared to meet inclusion criteria but were excluded (Item 16b). Usually this is detailed in a table in the Supplementary Files.

Data sharing:
10) The authors do not share the NMA data as no new raw data has been generated. However, the sharing of NMA data is encouraged. Additionally, the analysis code for the NMAs should be shared for transparency and reproducibility purposes.

Specific minor comments seeking clarity:
11) Line 65, Introduction: You mention evidence-based guidelines, but only cite trials or meta-analyses. Are there any clinical guidelines available on acne scarring?
12) Line 79, Introduction: Can you provide some examples of which treatments were not evaluated in previous NMAs?
13) Line 154, Results, Literature screening results: Please provide examples of what you mean by “did not have relevant data”. For example, did these articles not present the statistics required for the NMAs?
14) Line 158, Results, Literature screening results: What do the numbers “(17, 28-94)” represent?
15) Lines 166-173, NMA results, Network diagram: The description of the network diagram characteristics (e.g., the node size and line thickness) was already described in the Methods section. This description should be moved to the caption of the figures. In addition, please describe key features of the distribution of evidence in the network diagrams. For example, are most treatments compared to a particular control?
16) Line 245, Results: No results are presented for the sensitivity analyses (i.e., the leave-one-out analyses described in the Methods section). Did you conduct them?
17) Line 250, Discussion: What do the numbers “(95, 95)” represent?
18) Lines 250-252, Discussion: Please revise to clarify the contribution of this NMA. For example, “This NMA is the first to compare multiple domains of treatment types, including laser, drugs, and chemical peeling”.
19) Lines 320-321, Discussion: Please be more specific on how individual treatment plans can be developed based on the results of the NMA. The NMA assesses efficacy at the population level. Perhaps patients have different views of outcome importance, and that could help them decide at the individual level. You could also emphasize that NMA provides a coherent comparison of multiple treatment options, allowing them to rationally pick the treatment with the best evidence of its efficacy-safety profile.
20) Figures 5-6: Is it possible to add the actual SUCRA values beside the treatments in the legend?
21) Tables 1-7: In the captions, you need to describe the summary effect measure and credible interval presented (e.g., MD (95% CI)), how to interpret the direction of effect (e.g., do the summary effect measures compare the treatment in the row vs. treatment in the column, do values < 0 favor row or column treatment?), and what the different colours represent.

Suggested editorial revisions:
22) Line 21, Abstract: “nefigtwork” should be “network”.
23) Line 23, Abstract: Define the ECCA abbreviation.
24) Line 23, Abstract: Be consistent with the Goodman Baron abbreviation. It is defined to be “G&B”, but it is later reported as “GBS”.
25) Line 28, Abstract: Define the SUCRA abbreviation.
26) Lines 45-46, Introduction: Suggest revising “effective approaches for acne scarring” to “effective treatments for acne scarring”.
27) Line 80, Introduction: Suggest revising “this study was to” to “this study was conducted to”.
28) Line 106, Materials & Methods, Inclusion and exclusion criteria: Suggest revising “Articles were excluded for meta-analysis, …” to “Articles were excluded if they were a meta-analysis, …”.
29) Lines 110 and 115, Materials & Methods, Inclusion and exclusion criteria: When stating author initials, please follow the convention of stating the first initial of first name, followed by first initial of surname, to avoid confusion.
30) Line 128, Materials & Methods, Data analysis: I am not sure what “annealings” means. Do you mean to say “burn-in”?
31) Line 132, Materials & Methods, Data analysis: The citation of (Chaimani et al. 2013) would be better placed in the next sentence which talks about plots.
32) Line 134, Materials & Methods, Data analysis: Suggest revising “the size represents the entire sample size” to “the node size is proportional to the number of patients who received that intervention”.
33) Lines 135-136, Materials & Methods, Data analysis: Suggest revising “their width corresponds to the number of trials” to “their width is proportional to the number of trials making that comparison”.
34) Line 137, Materials & Methods, Data analysis: Define SUCRA abbreviation.
35) Line 178, Materials & Methods, ECCA scores: Define CI abbreviation.
36) Line 185, Materials & Methods, ECCA scores: I think you are referring to Fig5A, not Fig4A.

Experimental design

1) The key assumption of NMA is exchangeability. So, there should be no systematic differences between the studies’ characteristics which may modify the relative effects. Inconsistency checks, which were performed in this paper (i.e., node-splitting), are one way of checking this. In addition, there should be a qualitative assessment of how the studies compare in terms of effect modifiers. For example, how did the patient populations compare across studies in terms of scar severity? Were the treatments delivered in a sufficiently similar way across studies in terms of dose, frequency, and duration?

The description of methods also needs to be improved for transparency and reproducibility purposes. More specifically:
2) For each outcome, explicitly state which model was used. For example, data on continuous outcomes (e.g., ECCA, GBS, VAS) were likely synthesized with a Bayesian NMA model with a normal likelihood and identity link, while data on binary outcomes (e.g., patient satisfaction, AEs) were likely synthesized with a Bayesian NMA model with a binomial likelihood and log (or logit) link.
3) Please confirm that data on continuous outcomes were synthesized as mean differences (MD) or standardized mean differences (SMD). The Methods suggest SMDs, but the Results section suggest MDs.
4) If continuous data are synthesized as SMDs, justify this – was an outcome measured on different scales?
5) Please confirm if data on the binary outcomes were synthesized as relative risks (RRs) or logRRs.
6) Please justify why data on binary outcomes were synthesized as (log)RRs rather than (log) odds ratios.
7) State how convergence was assessed (e.g., did you inspect how well the chains mixed via trace plots).
8) State if both fixed and random effects models were fitted, and on what basis did you select a fixed effects model over a random effects model (or vice versa).
9) State how model fit was assessed.
10) State the specific software used to conduct the analyses. For example, was the gemtc package in R used?

The presentation of NMA results can also improve to assess the robustness of the NMA results. The following information should be added for each outcome:
11) A statement on whether the models successfully converged.
12) The model fit (e.g., the posterior residual deviance vs. the number of data points).
13) Detailed node-splitting results (i.e., the forest plots of the direct and indirect estimates), which may be presented in the supplementary files.
14) A note on which studies, if any, had to be excluded because they were not connected to the network.

Validity of the findings

There is not enough detail reported in the Methods and Results to adequately assess the robustness of the NMA results. Once the other comments are addressed, this may be assessed.

Additional comments

This systematic review and network meta-analysis (NMA) addresses a pressing issue in acne care, i.e., the treatment of acne scarring. The search strategy is strong, resulting in a large set of randomized trials examining a comprehensive set of treatments that has not been previously summarized in any known network meta-analyses. The manuscript could be significantly strengthened by providing more information to enable assessment of the robustness of the results, as well as reproducibility.

---

## Round 0.2 · Major Revisions

I thank to authors for their effort to improve the Methods section and share the NMA data. However, some further improvements are still needed, as detailed in the reviewer's comments, and I encourage the authors to ensure that the updated results have been consistently incorporated throughout the manuscript.

Reviewer 3 ·

Basic reporting

There are some areas where the spelling and grammar could be improved. Here are some suggestions (note this list is not comprehensive and authors should conduct a second check as well):

1. Page 1, line 2, Title: “…a systematic review and network meta-analysis”
2. Page 1, line 20, Abstract, Background: “This study aimed to assess…”
3. Page 1, lines 23-24, Abstract, Methods: “Outcomes included Echelle…”
4. Page 1, line 25, Abstract, Methods: “Bayesian network meta-analyses were…”
5. Page 2, lines 63-64, Introduction: “however, periodic reapplication may be required to maintain improved appearance.”
6. Page 2, line 85, Introduction: “these studies such as filler injections…”
7. Page 3, line 89, Introduction: “This study aimed to systematically review…”
8. Page 3, line 96, Materials & Methods: “An NMA was conducted according to a predefined protocol and reported following the PRISMA 2020 guidelines…”
9. Page 3, line 112, Inclusion and exclusion criteria: “adverse events”
10. Page 3, lines 120-124, Literature selection and data extraction: “Articles included at the title and abstract stage were further screened at the full text level against the inclusion and exclusion criteria, and those that were included were considered for the NMAs. Any disagreements were resolved through discussion or by seeking advice from a third researcher (XY, Z)”.
11. Page 5, line 183, Literature screening results: “65 articles did not report the data required for NMA…”.
12. Page 5, line 199, Network diagram: “The evidence network diagrams are shown…”

Suggestions for improving reporting in Abstract:

13. Page 1, line 25, Methods: Replace “VAS” with “pain”.

Suggestions for improving reporting in Introduction:

14. Page 2, lines 59-60: Please clarify what outcomes microneedling and PRP are efficacious in (e.g., reducing severity of acne scars).
15. Page 2, line 61: Please clarify what treatment(s) microneedling + PRP is superior to.
16. Page 2, lines 68-69: Suggest clarifying you only uncovered one guideline. For example: “Despite many treatment options available for acne scarring, there is a dearth of comprehensive evidence-based guidelines to help clinicians and patients choose an optimal treatment. To our knowledge, only one guideline specifically mentions treatments for acne scarring, although it primarily focuses on acne vulgaris management (Reynolds et al. 2024).”

Suggestions for improving reporting in Materials and Methods:

17. Page 3, line 100, Data sources and access: Suggest removing redundant sentence: “Data source is crucial for NMA.”
18. Page 4, lines 168-170, Data analysis: Clarify if the sensitivity analysis was done for every single study in the network, or only studies at high risk of bias.

Suggestions for improving reporting in Results:

19. Page 5, lines 199-202, Network diagram: “The results showed… using node-split analysis”. These sentences seem out of place. Suggest removing.
20. When presenting the CrIs in the text, please follow the convention (lower limit, upper limit) so that the negative signs are easier to read.
21. Page 6, line 216, ECCA scores: Remove “indicating the better efficacy of laser + PRP (p<0.05)”. “Statistical significance” (i.e., p < 0.05) should be avoided when interpreting results from a Bayesian analysis. The existing text preceding the listed MDs and 95% CrIs (“…ECCA scores were notably reduced after last combined with PRP for acne scarring compared with…”) suggests that laser + PRP has better efficacy than these treatments.
22. Page 6, lines 222-223, GBS: Remove “indicating no statistical significance (P>0.05)”.
23. Page 6, line 229, VAS scores: Suggest clarifying this is “Pain based on VAS scores”.
24. Page 6, line 232, VAS scores: Remove “indicating no statistical significance (P>0.05)”.
25. Page 6, line 233, VAS scores: Replace “GBS” with “VAS”.
26. Page 6, line 254, Erythema: Remove “indicating no statistical significance (P>0.05)”.
27. Page 7, line 263, Edema: Remove “indicating no statistical significance (P>0.05)”.
28. Page 7, lines 270-271, PIH: Remove “indicating no statistical significance (P>0.05)”.
29. Page 7, line 280, Publication bias: I think you should only refer to Figure 8.
30. Table 1: Please include the model fit statistics for both fixed and random effects models for each outcome.
31. Table 7: There are two cells with “lasers + chemical peeling” for edema. Double check labeling.
32. All figures and tables should define all abbreviations (including abbreviations in the captions and treatment names).

Suggestions for improving reporting in Discussion:

33. Page 7, lines 297-298: Clarify if this first sentence about the efficacy of laser + PRP and laser + filler injection is based on the NMA results, or external results. If external, provide a reference.
34. Page 9, lines 350-352: Provide references to justify the statements about the tolerability and short recovery associated with microneedling.

Experimental design

Regarding the research question:

1. Please clarify if you restricted the patient population to those who have recovered from acne and do not have new lesions. This could be better articulated in the title (e.g., “in patients with historic acne”), and in the last paragraph of the Introduction.

Some improvement in the description of Methods is required to provide clarity and meet what is typically expected in the NMA literature. Generally, when describing methods, it is better to state what an R function does, rather than name it. Here are some suggestions:

2. Page 1, lines 26-27, Abstract, Methods: “…performed using the gemtc package in R. Risk of bias was assessed using Cochrane Risk of Bias (RoB 2) tool, while publication bias was assessed via funnel plots.” No need to state software for these analyses.

3. Page 4, lines 140-153, Data analysis: Suggest revising this paragraph to remove mention of specific R commands. For example, (please confirm details and use appropriate references; also please see comment 12 regarding the using of I2 and revise accordingly): “Bayesian NMAs were performed using the gemtc package in R 4.4.1; a normal likelihood and identity link was applied to continuous outcomes, while a binomial likelihood and log link was applied to binary outcomes (insert reference for gemtc). An initial value of 2.5 was set and four chains were run, with the initial 10,000 iterations discarded as burn-in, and the results were summarized based on the subsequent 40,000 iterations for each chain with a thinning rate of 1. Convergence was visually assessed using Gelman and Rubin's shrink factor and quantitatively by the potential scale reduction factor (PSFR), where 1 ≤ PSRF < 1.05 was indicative of convergence (insert reference). Fixed and random effects models were fitted; the degree of heterogeneity was assessed through the I2 statistic. Results based on the random effects model were presented if I2≥50%. The goodness of fit was assessed using the mean sum of residual deviance to the number of data points ratio.”

4. Page 4, lines 140-153, Data analysis: The priors for the relative effects and between-study standard deviation (or variance) should be stated in this paragraph. If you let gemtc determine the priors, state this.

5. Page 4, line 145, Data analysis: What do you mean by “initial value of 2.5”? Usually, initial values are specified for all estimated parameters (e.g., the relative effects vs. the reference treatment and the between-study standard deviation if applicable). Please clarify in the text what parameter this initial value corresponds to.

6. Page 4, lines 155-159, Data analysis: The second sentence is a bit redundant; also note that we cannot conclude no inconsistency if p > 0.05 – we can only make conclusions if p < 0.05. Suggested revision: “The node splitting method (van Valkenhoef et al. 2016) was used to assess consistency in loops consisting of both direct and indirect evidence. A Bayesian p-value < 0.05 was indicative of inconsistency between the direct and indirect evidence.”

7. Page 4, lines 159-164, Data analysis: The description of the network diagrams should be presented at the beginning of this section (~line 138), as this is usually the first step to ensure connected networks. I also suggest revising these sentences to be: “Network diagrams were plotted to illustrate comparisons across treatments and to ensure connectivity. Each node represents an intervention, and the node size is proportional to the number of patients who received that intervention (Chaimani et al. 2013). Connection lines denote direct comparisons between two interventions, and their width is proportional to the number of trials making that comparison (Thom et al. 2019).”

8. Page 4, lines 164 – 167, Data analysis: The presentation of results can be described in a new paragraph. For example: “Synthesized effect sizes were presented as mean differences (MD) and relative risks (RR), depending on the type of outcome, along with the corresponding 95% credible intervals (CrI). The ranking of treatments was determined by calculating the Surface under the Cumulative Ranking curve to the total Area (SUCRA) for each intervention. SUCRA is the ratio of the surface under the cumulative ranking curve to the total area of the graph; larger values are indicative of more favorable outcomes (Salanti et al. 2011).” Note language such as “better intervention effect” is not recommended when only referencing ranking statistics.

9. Page 4, lines 170-171, Data analysis: Suggest noting the network package in STATA here: “Funnel plots were created using the network package in STATA 15.1 to assess publication bias (Egger et al. 1997).”

10. Page 5, lines 172-173, Data analysis: This sentence can be removed if the software are mentioned earlier after revisions: “All statistical analyses were performed using R 4.4.1 and STATA 15.1 software”.

11. Page 5, line 173: Since the NMAs were conducted in a Bayesian framework, we should avoid using statements regarding “statistical significance” which are typically reserved for frequentist analyses. Instead, you could say something like “Evidence supporting superior efficacy or safety was noted when the 95% CrIs excluded the null effect.”

Analysis procedures that could be improved:

12. Page 4, line 147-148, Data analysis: Classifying the degree of heterogeneity based on I2 is not recommended (see Borenstein 2024). Instead, the fit of the fixed and random effects models should be compared based on the posterior mean of total residual deviance (is it close to the number of data points?) or similarly, the mean sum of residual deviance to the number of data points ratio (is it close to 1?), along with the deviance information criterion (DIC; lower DIC values are preferred). The posterior distribution of the between-study standard deviation should also be inspected to assess whether it was sufficiently estimated; if not, a random effects model needs to be rerun with an informative prior on the between-study standard deviation (or variance).

Borenstein M. Avoiding common mistakes in meta-analysis: Understanding the distinct roles of Q, I-squared, tau-squared, and the prediction interval in reporting heterogeneity. Res Syn Meth. 2024; 15(2): 354-368. doi:10.1002/jrsm.1678

Validity of the findings

I am happy to see the authors have added some caution when interpreting SUCRA in the Results of the main paper; this needs to be implemented throughout the Abstract, Discussion, and Conclusion. The authors should be careful when declaring a treatment to have the “best efficacy” or “more effective” based on SUCRA. Instead, I recommend being transparent about the strength of evidence supporting a treatment’s superior efficacy or safety over another treatment (i.e., if the 95% CrI excludes the null effect) and using language such as “ranked best”. Here are some more specific suggestions (which can be edited and/or improved!):

1. Page 1, lines 30-34, Abstract, Results: “Laser + platelet-rich plasma (PRP) ranked best in reducing ECCA scores (surface under cumulative ranking curve to the total area [SUCRA]: 98.4%), laser + filler injection ranked best in reducing GBS (SUCRA: 90.2%), and laser + chemical peels ranked the best in patient satisfaction (SUCRA: 85.6%). Microneedling was ranked as the most tolerable in terms of pain (SUCRA: 99.9%).”

2. Page 1, line 34, Abstract, Results: Consider saying something about the AEs such as, “There was no strong evidence suggesting a treatment reduced the risk of erythema nor post-inflammatory hyperpigmentation compared to the other treatments.”

3. Page 1, lines 35-37, Abstract, Conclusions: “The evidence suggests laser combined with PRP or filler injections are the best options for reducing scar severity, while laser combined with chemical peeling yields the best patient satisfaction. Laser combined with other therapies should be considered to optimize treatment of acne scarring.”

4. Page 7, lines 290-293, Discussion: “Laser + PRP ranked best in reducing ECCA scores, and was notably more efficacious than microneedling alone, lasers alone, chemical peeling, PRP alone, microneedling + PRP, lasers + drugs, lasers + filler injections. Laser + filler injection ranked best in terms of improving GBS, but there was no evidence to conclude a notable difference in efficacy between treatments. Microneedling ranked best in terms of yielding the least pain measured by VAS scores, but there was no evidence to conclude a notable difference in tolerability between treatments. Laser + chemical peeling ranked best in patient satisfaction, and was notably more efficacious than microneedling alone, lasers alone, chemical peeling alone, and microneedling + filler injections.”

5. Page 7, lines 293-296, Discussion: Focusing on the AEs for laser + PRP and laser + chemical peeling is good, as these are the most promising treatments in reducing acne scars and improving patient satisfaction. Consider specifying which comparators they are worse than in terms of AEs (e.g., “The incidence of erythema is higher for laser + peeling than all other treatments with available data except for lasers alone, although there was not enough evidence to conclude a notable difference for any comparison.” ). Make a note if any “second best” treatments have a smaller incidence of AEs, if any.

6. Page 9, lines 383-387, Conclusion: “This NMA provides evidence suggesting that laser + PRP and laser + filler injection should be considered as optimal treatments in reducing acne scars, while noting that patients may be most satisfied following laser + chemical peeling treatment and microneedling alone may be the most tolerable in terms of pain.” Alternatively, you could say something more general such as, “Laser combined with other therapies should be considered to optimize treatment of acne scarring. More evidence is required to conclude definitive superiority of these treatments over all other regimens, and to better understand tolerability as well as the risks of adverse events.”

Other comments:

7. Page 1, lines 30-34, Abstract, Results: Please double-check the top treatments and their SUCRA values, as some of the current descriptions do not exactly match what is reported in Figure 5.

8. In the Discussion, please clarify if the treatments investigated in the NMA have been proven to be more efficacious than an appropriate “placebo” in randomized controlled trials not included in the network, or if their efficacy has been established through before/after studies.

9. Page 8, lines 309-310: Provide more justification about the additive effect of laser and PRP (for example, reference the relative effect of laser + PRP vs. laser alone and laser + PRP vs. PRP alone).

Additional comments

I appreciate the authors’ effort to improve the description of the Methods and commend them for sharing the NMA data. Some improvements are required, and I have provided specific examples. I encourage the authors to double check that the updated results have been incorporated throughout the manuscript.

---

## Round 0.3 · Minor Revisions

I thank to the authors for the revision; however, the paper needs some additional work. Please read the minor suggestions of the reviewer and upload the revised manuscript.

**PeerJ Staff Note**: Please ensure that all review, editorial, and staff comments are addressed in a response letter and that any edits or clarifications mentioned in the letter are also inserted into the revised manuscript where appropriate.

Reviewer 3 ·

Basic reporting

Line 93: Suggested revision: "More specifically, this NMA aimed to systematically..."

Lines 95-96: This NMA did not "provide recommendations for different types and severities of acne scarring". Either comment on this in the Discussion (you could say that limited evidence did not permit subgroup analyses, if this was indeed the case), or remove this from the objective.

Lines 165-166: Suggested revision: "...when both direct and indirect evidence was available for at least one comparison in the network."

Line 233: Suggested revision: "...as differences in tolerability among interventions."

Line 235-236: Please clarify what you mean by "and those with satisfaction were analyzed together". Do you mean "and this outcome was analyzed by comparing the proportions of patients that were satisfied or very satisfied and the proportion of patients that were not." ?

Line 267: Suggested revision: "... interpreted as differences in PIH incidence among..."

Line 368: Suggested revision: "...literature was not included in this NMA..."

Experimental design

No comment.

Validity of the findings

Lines 332 - 333: "Our NMA uncovered that microneedling alone was excellent in reducing pain and improving the appearance of scars and quality of life." This sentence needs to be revised as there were no significant differences between treatments for pain. For example, "microneedling alone may be the best in terms of pain tolerability". I don't think there was an evidence suggesting it was the best for improving the appearance of scars and quality of life, so this should be deleted.

Additional comments

The authors have done a good job incorporating my previous comments and acknowledging the limitations of the NMA results. I have made some very minor suggestions for the authors to consider.

---

## Round 0.4 · accepted · Accept

I thank to the authors for making revisions.

Reviewer 3 ·

Basic reporting

No comment.

Experimental design

No comment.

Validity of the findings

No comment.

Additional comments

No comment.